# ZERO-SHOT REINFORCEMENT LEARNING FROM LOW QUALITY DATA

## ABSTRACT

Zero-shot reinforcement learning (RL) promises to provide agents that can perform *any* task in an environment after an offline pre-training phase. *Forward-backward* (FB) representations represent remarkable progress towards this ideal, achieving 85% of the performance of task-specific agents in this setting. However, such performance is contingent on access to large and diverse datasets for pre-training, which cannot be expected for most real problems. Here, we explore how FB performance degrades when trained on small datasets that lack diversity, and mitigate it with *conservatism*, a well-established feature of performant offline RL algorithms. We evaluate our family of methods across various datasets, domains and tasks, reaching 150% of vanilla FB performance in aggregate. Somewhat surprisingly, conservative FB algorithms also outperform the task-specific baseline, despite lacking access to reward labels and being required to maintain policies for all tasks. Conservative FB algorithms perform no worse than vanilla FB on full datasets, and so present little downside over their predecessor. Our code is available open-source via https://anonymous.4open.science/r/conservative-world-models-4903.

## 1 INTRODUCTION

Today's large pre-trained models exhibit an impressive ability to generalise to unseen vision (Rombach et al., 2022) and language (Brown et al., 2020) tasks. Zero-shot reinforcement learning (RL) methods leveraging successor features (SFs) (Barreto et al., 2017; Borsa et al., 2018) and forward-backward (FB) representations (Touati & Ollivier, 2021) aim to instantiate a similar idea in the sequential decision-making context. Recently, FB representations in particular have been shown to perform zero-shot RL remarkably well: provided a dataset of reward-free transitions from a target environment, FB can return policies for *any* task in the environment that are 85% as performant as those returned by offline RL algorithms explicitly trained for each task. This is achieved with no prior knowledge of the tasks, zero planning, and no online interaction.

However, such performance is only achievable if the pre-training dataset is large and diverse. Real datasets, like those produced by an existing controller or collected by a task-directed agent, are usually small and lack diversity. Even if we design agents to exhaustively explore environments, as is done in Unsupervised RL (Jaderberg et al., 2016), they suffer the impracticalities of the online RL algorithms we are trying to avoid: they act dangerously in safety-critical environments, and data collection can be time-consuming.

Is it possible to relax this requirement and perform zero-shot RL using more realistic datasets? This is the primary question we address in this paper. We begin by establishing that current methods suffer in this regime because they overestimate the value of out-of-distribution state-action pairs. In response, we adapt ideas from *conservatism* in offline RL (Kumar et al., 2020) for use with FB representations, creating two new algorithms: *value-conservative FB representations* (VC-FB) (Figure 1 (*right*)) and *measure-conservative FB representations* (MC-FB). The former regularises the predicted value of out-of-distribution state-action pairs, whilst the latter regularises future state visitation measures. In experiments across varied domains, tasks and datasets, we show our proposals outperform FB and SF-based approaches by up to 150% in aggregate, and even surpass a task-specific baseline. Finally, we establish that both VC-FB and MC-FB perform no worse than FB on large datasets, and so present little downside over their predecessor.

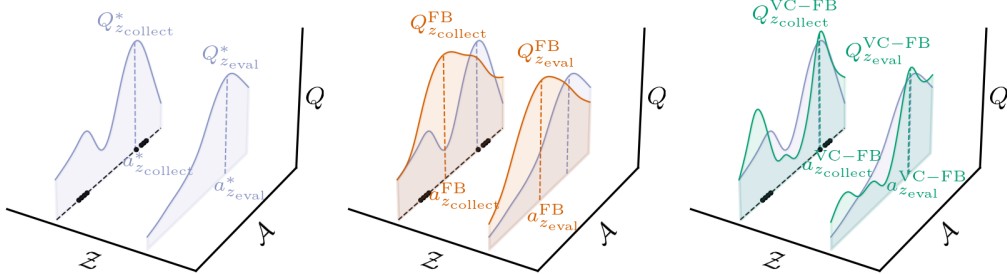

Figure 1: **Toy illustration of FB's failure mode on sub-optimal datasets and VC-FB's resolution.**. (*Left*) Zero-shot RL methods train on a dataset collected by a behaviour policy optimising against task $z_{\text{collect}}$, yet generalise to new tasks $z_{\text{eval}}$. Both tasks have associated optimal value functions $Q^*_{z_{\text{collect}}}$ and $Q^*_{z_{\text{eval}}}$ for a given marginal state. (*Middle*) Forward-backward (FB) representations overestimate the value of actions not in the dataset for all tasks. (*Right*) Value-conservative forward-backward (VC-FB) representations suppress the value of actions not in the dataset for all tasks. Black dots represent state-action samples present in the dataset.

## 2 BACKGROUND

**Problem formulation.** The zero-shot RL problem extends the standard RL setup of a Markov decision process (MDP) (Sutton & Barto, 2018). We consider the class of continuous, finite-horizon MDPs, in which $\mathcal{S} \in \mathbb{R}^n$ and $\mathcal{A} \in \mathbb{R}^m$ are continuous spaces of environment states and agent actions and $\mathcal{P} : \mathcal{S} \times \mathcal{A} \rightarrow \Delta(\mathcal{S})$ is a stochastic state transition function (Bellman, 1957). At each timestep $t$, the agent observes state $s_t$, selects action $a_t$ according to a policy function $\pi$ and transitions to the next state $s_{t+1} \sim \mathcal{P}(\cdot|s_t, a_t)$. This process repeats until a terminal timestep $t = T$. The MDP formalism is completed by reward function $\mathcal{R} : \mathcal{S} \rightarrow \mathbb{R}_{\geq 0}$, which maps states to non-negative rewards[1], and a discount factor $\gamma \in [0, 1]$. Any given $\mathcal{R}$ instantiates a *task* for the agent, namely to maximise the expected discounted sum of rewards for visited states, $\mathbb{E}_{\pi,\mathcal{P}} \sum_{t=0}^{T-1} \gamma^t \mathcal{R}(s_{t+1})$. In contrast with standard RL, which considers only a single $\mathcal{R}$, we are interested in agents that can solve *any* arbitrary task in an environment, each characterised by a different reward function but otherwise sharing a common MDP structure $(\mathcal{S}, \mathcal{A}, \mathcal{P}, T, \gamma)$ (Borsa et al., 2018). During a pre-training phase, we give the agent access to a static dataset of reward-free transitions $\mathcal{D} = \{(s_i, a_i, s_{i+1})\}_{i\in\{1,...,k\}}$ generated by an unknown behaviour policy. Once a task is revealed downstream, the agent must return a good policy for that task with no further planning or learning.

**Forward-backward representations.** FB representations tackle the zero-shot RL problem using *successor measures*, which generalise Dayan (1993)'s successor representations to continuous state spaces (Blier et al., 2021). A successor measure gives the expected discounted time spent in each subset of states $S_+ \subset \mathcal{S}$ after starting in state $s_0$, taking action $a_0$, and following policy $\pi$ thereafter:

$$M^\pi(s_0, a_0, S_+) := \sum_{t=0}^{T-1} \gamma^t \Pr(s_{t+1} \in S_+|(s_0, a_0), \pi), \; \forall \, S_+ \subset \mathcal{S}. \tag{1}$$

The successor measure is task-independent, but for any given task (with reward function $\mathcal{R}$), the state-action value ($Q$) function is the integral of $\mathcal{R}$ with respect to $M^\pi$:

$$Q^\pi_\mathcal{R}(s_0, a_0) := \int_{s_+ \in \mathcal{S}} \mathcal{R}(s_+) M^\pi(s_0, a_0, ds_+). \tag{2}$$

$\pi$ is an optimal policy for $\mathcal{R}$ if it maximises its own $Q$-function, i.e. $\pi(s) = \text{argmax}_a Q^\pi_\mathcal{R}(s, a), \forall s$.

FB representations approximate the successor measures of near-optimal policies for an infinite family of tasks. The key idea is to parameterise this family by a distribution of real-valued task vectors $\mathcal{Z} \in \Delta(\mathbb{R}^d)$. For any given $z \sim \mathcal{Z}$, let $\pi_z = \pi(s, z)$ denote the associated parameterised policy for that task. The first component of an FB representation is the *forward* model $F : \mathcal{S} \times \mathcal{A} \times \mathbb{R}^d \rightarrow \mathbb{R}^d$, which takes in a state $s_0 \in \mathcal{S}$, action $a_0 \in \mathcal{A}$ and task vector $z$ and outputs an embedding vector

---

[1]In more general formulations, rewards can be negative and dependent on (state, action, next state) triplets, but we consider this special case here.

(also in $\mathbb{R}^d$), which can be intuitively understood as summarising the distribution of future states visited after taking $a_0$ in $s_0$ and following $\pi_z$ thereafter. The second component is a *backward* model $B : \mathcal{S} \to \mathbb{R}^d$, which outputs another embedding vector summarising the distribution of states visited before a given state $s_+ \in \mathcal{S}$ (it is not conditioned on $z$) . Touati & Ollivier (2021) show that $F$ and $B$ can be combined to form a rank-$d$ approximation to the successor measure for any policy $\pi_z$:

$$M^{\pi_z}(s_0, a_0, \mathrm{d}s_+) \approx F(s_0, a_0, z)^\top B(s_+)\rho(\mathrm{d}s_+), \ \forall \ s_+ \in \mathcal{S}. \tag{3}$$

$\rho$ is a marginal state distribution, which in practice is that of the pre-training dataset $\mathcal{D}$. Intuitively, Equation 3 says that the approximated successor measure under $\pi_z$ from $(s_0, a_0)$ to $s_+$ is high if their respective forward and backward embeddings are similar (i.e. large dot product).

In turn, by Equation 2, an FB representation can be used to approximate the $Q$ function of $\pi_z$ with respect to any reward function $\mathcal{R}$ as follows:

$$Q_{\mathcal{R}}^{\pi_z}(s_0, a_0) \approx \int_{s_+ \in \mathcal{S}} \mathcal{R}(s_+)F(s_0, a_0, z)^\top B(s_+)\rho(\mathrm{d}s_+) = F(s_0, a_0, z)^\top \mathbb{E}_{s_+ \sim \rho}[\ \mathcal{R}(s_+)B(s_+)\ ]. \tag{4}$$

**Pre-training FB.** Since the successor measure satisfies a Bellman equation (Blier et al., 2021), $F$ and $B$ can be pre-trained to improve the approximation in Equation 3 by performing temporal difference (TD) updates (Samuel, 1959; Sutton, 1988) using transition data sampled from $\mathcal{D}$:

$$\mathcal{L}_{\text{FB}} = \mathbb{E}_{(s_t, a_t, s_{t+1}, s_+) \sim \mathcal{D}, z \sim \mathcal{Z}}[\left(F(s_t, a_t, z)^\top B(s_+) - \gamma \bar{F}(s_{t+1}, \pi_z(s_{t+1}), z)^\top \bar{B}(s_+)\right)^2$$
$$- 2F(s_t, a_t, z)^\top B(s_{t+1})], \quad (5)$$

where $s_+$ is sampled independently of $(s_t, a_t, s_{t+1})$ and $\bar{F}$ and $\bar{B}$ are lagging target networks. See Touati & Ollivier (2021) for a full derivation of this TD update, and our Appendix B.1 for practical implementation details including the specific choice of task sampling distribution $\mathcal{Z}$.

**Using FB for zero-shot RL.** Touati et al. (2023) show that using FB for zero-shot RL begins with defining the parameterised policy family as:

$$\pi_z(s) = \text{argmax}_a F(s, a, z)^\top z. \tag{6}$$

Then, relating Equations 4 and 6, we find $z = \mathbb{E}_{s_+ \sim \rho}[\ \mathcal{R}(s_+)B(s_+)\ ]$ for some reward function $\mathcal{R}$. If $z$ lies within the task sampling distribution $\mathcal{Z}$ used during pre-training, then $\pi_z(s) \approx \text{argmax}_a Q_{\mathcal{R}}^{\pi_z}(s, a)$, and hence this policy is approximately optimal for $\mathcal{R}$. In practice, continuous action spaces necessitate learning an approximation to the argmax in Equation 6 via a task-conditioned policy model,[2] but the optimality relationship continues to approximately hold. We can thus exploit it to obtain the following two-step process for performing zero-shot RL:

1. Provided access to a dataset $\mathcal{D}_{\text{labelled}}$ of states distributed as $\rho$ labelled with rewards by a target reward function $\mathcal{R}^*$, estimate $z^* \approx \mathbb{E}_{(s, r^*) \sim \mathcal{D}_{\text{labelled}}}[\ r^*B(s)\ ]$ by simple averaging. For a goal-reaching task with goal $s_g$, define the task vector directly as $z^* = B(s_g)$.

2. In theory, a near-optimal policy $\pi_{z^*}$ is given analytically via Equation 6. In practice, obtain the policy by passing $z^*$ as a parameter to the task-conditioned policy model.

Since this process requires no further planning or learning, the goal of zero-shot RL is realised.

Alternative zero-shot RL methods utilise *successor features* (SF) (Borsa et al., 2018). The value-space algorithms we propose next are fully-compatible with SF, as derived in Appendix E, but we focus our analysis on FB because of its superior empirical performance (Touati et al., 2023).

## 3 CONSERVATIVE FORWARD-BACKWARD REPRESENTATIONS

We begin by examining the FB loss (Equation 5) more closely. The TD target includes an action produced by the current policy $a_{t+1} = \pi_z(s_{t+1})$. Equation 6 shows that this action is the current best estimate of the optimal action in state $s$ for task $z$. When training on a finite dataset, this maximisation does not constrain the policy to actions observed in the dataset, and so the policy can become biased towards out-of-distribution (OOD) actions thought to be of high value–a well-observed

---

[2]This model is learnt concurrently to the FB representation itself in an actor-critic formulation (Lillicrap et al., 2016), as per the algorithm in Appendix B.1.5.

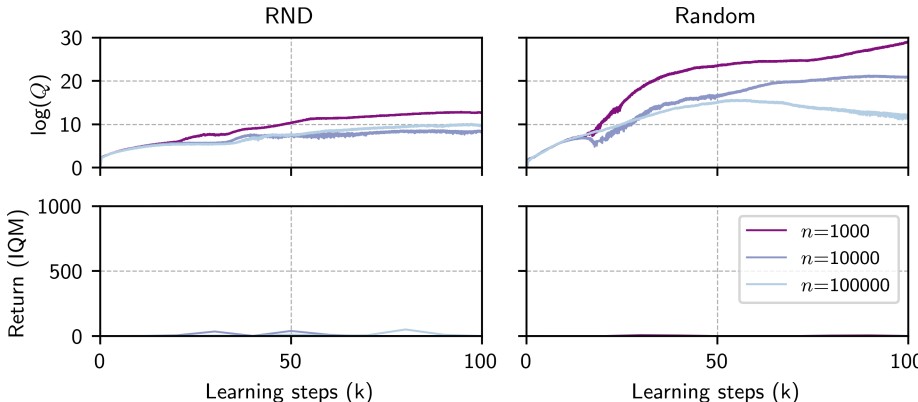

Figure 2: **FB value overestimation with respect to dataset size $n$ and quality.** Log $Q$ values and IQM of rollout performance on all Point-mass Maze tasks for datasets RND and RANDOM. $Q$ values predicted during training increase as both the size and "quality" of the dataset decrease. This contradicts the low return of all resultant policies. Informally, we say the RND dataset is "high" quality, and the RANDOM dataset is "low" quality–see Appendix A.2 for more details.

phenomenon in offline RL (Kumar et al., 2019a; 2020). In such instances, the TD targets may be evaluated at state-action pairs outside the dataset, making them unreliable and causing errors in the measure and value predictions. Figure 2 shows the overestimation of $Q$ as dataset size and quality is varied. The smaller and less diverse the dataset, the more $Q$ values tend to be overestimated.

The canonical fix for value overestimation in offline RL is conservative $Q$-learning (CQL) (Kumar et al., 2019a; 2020). Intuitively, CQL suppresses the values of OOD actions to be below those of in-distribution actions, and so approximately constrains the agent's policy to actions observed in the dataset. To achieve this, a new term is added to the usual $Q$ loss function

$$\mathcal{L}_{\text{CQL}} = \alpha \cdot \left( \mathbb{E}_{s \sim \mathcal{D}, a \sim \mu(a|s)}[Q(s,a)] - \mathbb{E}_{(s,a) \sim \mathcal{D}}[Q(s,a)] - \mathcal{H}(\mu) \right) + \mathcal{L}_{\text{Q}}, \qquad (7)$$

where $\alpha$ is a scaling parameter, $\mu(a|s)$ is a policy distribution selected to find the maximum value of the current $Q$ function iterate, $\mathcal{H}(\mu)$ is the entropy of $\mu$ used for regularisation, and $\mathcal{L}_{\text{Q}}$ is the normal TD loss on $Q$. Equation 7 has the dual effect of minimising the peaks in $Q$ under $\mu$ whilst maximising $Q$ for state-action pairs in the dataset. This proves to be a useful inductive bias, mitigating value overestimation and producing state-of-the-art results on offline RL benchmarks (Fu et al., 2020).

We can replicate a similar inductive bias in the FB context, substituting $F(s,a,z)^{\top}z$ for $Q$ in Equation 7 and adding the normal FB loss (Equation 5)

$$\mathcal{L}_{\text{VC-FB}} = \alpha \cdot \left( \mathbb{E}_{s \sim \mathcal{D}, a \sim \mu(a|s), z \sim \mathcal{Z}}[F(s,a,z)^{\top}z] - \mathbb{E}_{(s,a) \sim \mathcal{D}, z \sim \mathcal{Z}}[F(s,a,z)^{\top}z] - \mathcal{H}(\mu) \right) + \mathcal{L}_{\text{FB}}.$$
$$(8)$$

The key difference between Equations 7 and 8 is that the former suppresses the value of OOD actions for one task, whereas the latter does so for all task vectors drawn from $\mathcal{Z}$.[3] We discuss the usefulness of this inductive bias in Section 3.1. We call models learnt with this loss *value-conservative forward-backward representations* (VC-FB).

Because FB derives $Q$ functions from successor measures (Equation 4), and because (by assumption) rewards are non-negative, suppressing the predicted measures for OOD actions provides an alternative route to suppressing their $Q$ values. As we did with VC-FB, we can substitute FB's successor measure approximation $F(s,a,z)^{\top}B(s_+)$ into Equation 7, which yields:

$$\mathcal{L}_{\text{MC-FB}} = \alpha \cdot \left( \mathbb{E}_{s \sim \mathcal{D}, a \sim \mu(a|s), z \sim \mathcal{Z}, s_+ \sim \mathcal{D}}[F(s,a,z)^{\top}B(s_+)] \right.$$
$$\left. - \mathbb{E}_{(s,a) \sim \mathcal{D}, z \sim \mathcal{Z}, s_+ \sim \mathcal{D}}[F(s,a,z)^{\top}B(s_+)] - \mathcal{H}(\mu) \right) + \mathcal{L}_{\text{FB}}. \quad (9)$$

---

[3]An intuitively appealing alternative, given some a priori knowledge of the downstream tasks for which the model is to be used, would be to bias the sampling of task vectors $z$ used in the conservativism penalty towards those derived from plausible tasks $\mathcal{R}^*$ via the backward model, i.e. $z = \mathbb{E}_{s \sim \mathcal{D}}[\mathcal{R}^*(s)B(s)]$. We consider one instantiation of this *directed* conservatism approach in Appendix B.1.6, but find that it generally performs worse than undirected sampling in our experimental settings.

(a)          (b)          (c)

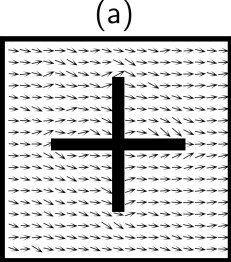 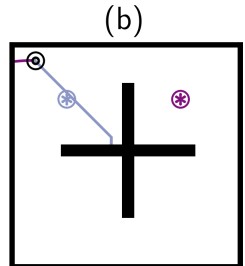 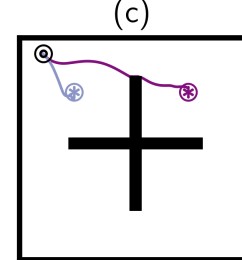

Figure 3: **Ignoring out-of-distribution actions.**. The agents are tasked with learning separate policies for reaching ⊛ and ⊛. (a) RND dataset with all "left" actions removed; quivers represent the mean action direction in each state bin. (b) Best FB rollout after 1 million learning steps. (c) Best VC-FB performance after 1 million learning steps. FB overestimates the value of OOD actions and cannot complete either task; VC-FB synthesises the requisite information from the dataset and completes both tasks.

Equation 9 has the effect of suppressing the expected visitation count to goal state $s_+$ when taking an OOD action for all task vectors drawn from $\mathcal{Z}$. As such, we call this variant a *measure-conservative forward-backward representation* (MC-FB). Since it is not obvious *a priori* whether the VC-FB or MC-FB form of conservatism would be more effective in practice, we evaluate both in Section 4.

Implementing conservative FB representations requires two new model components: 1) a conservative penalty scaling factor $\alpha$ and 2) a way of obtaining policy distribution $\mu(a|s)$ that maximises the current $Q$ function iterate. For 1), we observe fixed values of $\alpha$ leading to fragile performance, so dynamically tune it at each learning step using Lagrangian dual-gradient descent as per Kumar et al. (2020). Appendix B.1.4 discusses this procedure in more detail. For 2), the choice of maximum entropy regularisation following Kumar et al. (2020)'s CQL($\mathcal{H}$) allows $\mu$ to be approximated conveniently with a log-sum exponential across $Q$ values derived from the current policy distribution and a uniform distribution. That this is true is not obvious, so we refer the reader to the detail and derivations in Section 3.2, Appendix A, and Appendix E of Kumar et al. (2020), as well as our adjustments to Kumar et al. (2020)'s theory in Appendix B.1.3. Code snippets demonstrating the required changes to a vanilla FB implementation are provided in Appendix I. We emphasise these additions represent only a small increase in the number of lines required to implement FB.

## 3.1 A DIDACTIC EXAMPLE

To understand situations in which a conservative zero-shot RL methods may be useful, we introduce a modified version of Point-mass Maze from the ExORL benchmark (Yarats et al., 2022). Episodes begin with a point-mass initialised in the upper left of the maze (◎), and the agent is tasked with selecting $x$ and $y$ tilt directions such that the mass is moved towards one of two goal locations (⊛ and ⊛). The action space is two-dimensional and bounded in $[-1, 1]$. We take the RND dataset and remove all "left" actions such that $a_x \in [0, 1]$ and $a_y \in [-1, 1]$, creating a dataset that has the necessary information for solving the tasks, but is inexhaustive (Figure 3 (a)). We train FB and VC-FB on this dataset and plot the highest-reward trajectories–Figure 3 (b) and (c). FB overestimates the value of OOD actions and cannot complete either task. Conversely, VC-FB synthesises the requisite information from the dataset and completes both tasks.

The above example is engineered for exposition, but we expect conservatism to be helpful in more general contexts. Low-value actions for one task can often be low value for other tasks and, importantly, the more performant the behaviour policy, the less likely such low value actions are to be in the dataset. Consider the four tasks in the Walker environment: {walk, stand, run, flip}, where all tasks require the robot to stand from a seated position before exemplifying different behaviours. If the dataset includes actions that are antithetical to standing, as might be the case if the behaviour policy used to collect the dataset is highly exploratory, then both FB and VC-FB can observe their low value across tasks. However, if the dataset does not include such actions, as might be the case if it was collected via a near-optimal controller that never fails to stand, then FB may overestimate the value of not standing across tasks, and VC-FB would correctly devalue them. We extend these observations to more varied environments in the section that follows.

## 4 EXPERIMENTS

In this section we perform an empirical study to evaluate our proposals. We seek answers to three questions: **(Q1)** Can our proposals from Section 3 improve FB performance on small and/or low-quality datasets? **(Q2)** How does the performance of VC-FB and MC-FB vary with respect to task type and dataset diversity? **(Q3)** Do we sacrifice performance on full datasets for performance on small and/or low-quality datasets?

### 4.1 SETUP

We respond to these questions using the ExORL benchmark, which provides datasets collected by unsupervised exploratory algorithms on the DeepMind Control Suite (Yarats et al., 2022; Tassa et al., 2018). We select three of the same domains as Touati & Ollivier (2021): Walker, Quadruped and Point-mass Maze, but substitute Jaco for Cheetah. This provides two locomotion domains and two goal-reaching domains. Within each domain, we evaluate on all tasks provided by the DeepMind Control Suite for a total of 17 tasks across four domains. Full details are provided in Appendix A.1.

We pre-train on three datasets of varying quality. Although there is no unambiguous metric for quantifying dataset quality, we use the reported performance of offline TD3 on Point-mass Maze for each dataset as a proxy. We choose datasets collected via Random Network Distillation (RND) (Burda et al., 2018), Diversity is All You Need (DIAYN) (Eysenbach et al., 2018), and RANDOM policies, where agents trained on RND are the most performant, on DIAYN are median performers, and on RANDOM are the least performant. As well as selecting for quality, we also select for size. The ExORL datasets have up to 10 million transitions per domain. We uniformly sub-sample 100,000 transitions from these to create datasets that may be considered more realistically sized for real-world applications. More details on the datasets are provided in Appendix A.2, which includes a visualisation of the state coverage for each dataset on Point-mass Maze (Figure 6).

### 4.2 BASELINES

We use FB and SF with features from Laplacian eigenfunctions (SF-LAP) as our zero-shot RL baselines–the two most performant methods in Touati et al. (2023). As single-task RL baselines, we use CQL and offline TD3 trained on the same datasets relabelled with task rewards. CQL approximates what an algorithm with similar mechanistics can achieve when optimising for one task in a domain rather than all tasks. Offline TD3 exhibits the best aggregate single-task performance on the ExORL benchmark, so it should be indicative of the maximum performance we could expect to extract from a dataset. Full implementation details for all algorithms are provided in Appendix B.

We evaluate the cumulative reward (hereafter called score) achieved by VC-FB, MC-FB and our baselines on each task across five random seeds. To mitigate the well-established pitfalls of stochastic RL algorithm evaluation, we employ the best practice recommendations of Agarwal et al. (2021) when reporting task scores. Concretely, we run each algorithm for 1 million learning steps, evaluating task scores at checkpoints of 20,000 steps. At each checkpoint, we perform 10 rollouts, record the score of each, and find the interquartile mean (IQM). We average across seeds at each checkpoint to create the learning curves reported in Appendix H. From each learning curve, we extract task scores from the learning step for which the all-task IQM is maximised across seeds. Results are reported with 95% confidence intervals obtained via stratified bootstrapping (Efron, 1992). Aggregation across tasks, domains and datasets is always performed by evaluating the IQM. Full implementation details are provided in Appendix B.1.

### 4.3 RESULTS

**Q1.** We report the aggregate performance of all zero-shot RL methods and CQL in Figure 4. Both MC-FB and VC-FB outperform FB, achieving **150%** and **137%** FB performance respectively. The performance gap between FB and SF-LAP is consistent with the results in Touati et al. (2023). MC-FB and VC-FB outperform our single-task baseline in expectation, reaching 111% and 120% of CQL performance respectively *despite not having access to task-specific reward labels and needing to fit policies for all tasks*. This is a surprising result, and to the best of our knowledge, the first time a multi-task offline agent has been shown to outperform a single-task analogue. CQL outperforms

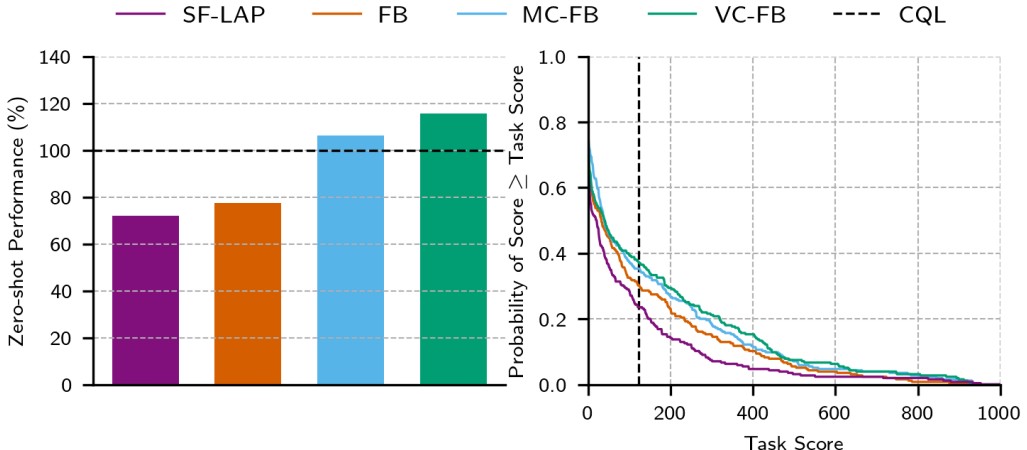

Figure 4: **Aggregate zero-shot performance.** *(Left)* IQM of task scores across datasets and domains, normalised against the performance of CQL, our baseline. *(Right)* Performance profiles showing the distribution of scores across all tasks and domains. Both conservative FB variants stochastically dominate vanilla FB–see Agarwal et al. (2021) for performance profile exposition. The black dashed line represents the IQM of CQL performance across all datasets, domains, tasks and seeds.

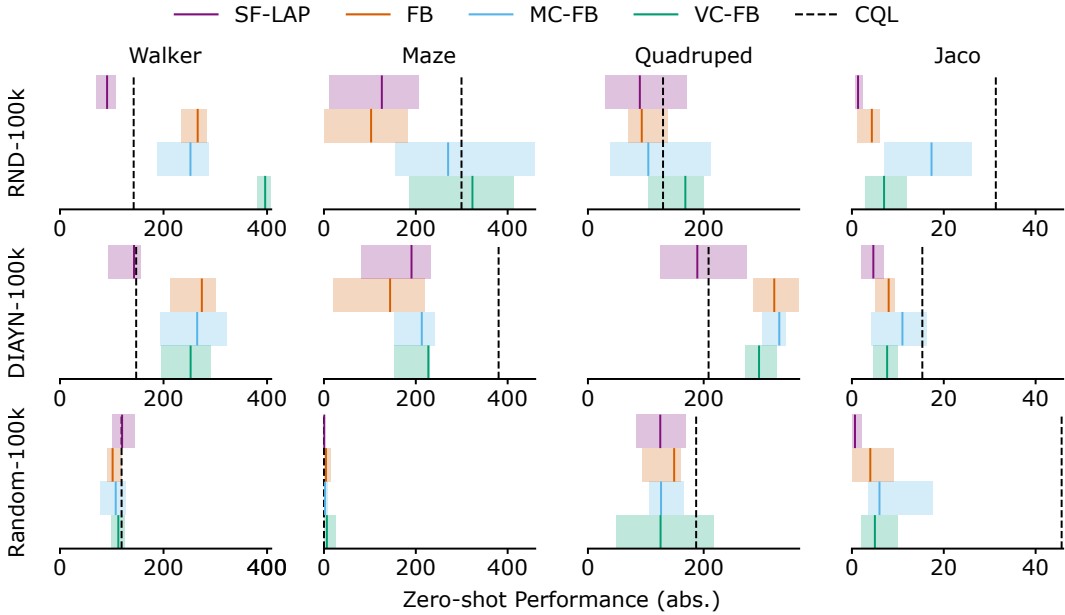

Figure 5: **Performance by dataset and domain.** IQM scores across tasks/seeds with 95% confidence intervals.

offline TD3 in aggregate, so we drop offline TD3 from the core analysis, but report its full results in Appendix C alongside all other methods. We note FB achieves 80% of single-task offline TD3, which roughly aligns with the 85% performance on the full datasets reported by Touati et al. (2023). *D*VC-FB (footnote 2; Appendix B.1.6) results are reported in Appendix C; it improves on vanilla FB by 9% in aggregate, but is outperformed by both VC-FB (37%) and MC-FB (26%). Reasons for these discrepancies are provided in Section 5.

**Q2.** We decompose the methods' performance with respect to domain and dataset diversity in Figure 5. The largest gap in performance between the conservative FB variants and FB is on RND, the highest-quality dataset. VC-FB and MC-FB reach 253% and 184% of FB performance respectively, and outperform CQL on three of the four domains. On DIAYN, the conservative variants

Table 1: **Aggregated performance on full datasets.** IQM scores aggregated over domains and tasks for all datasets, averaged across three seeds. Both VC-FB and MC-FB maintain the performance of FB; the largest relative performance improvement is on RANDOM.

| Dataset | Domain | Task | FB | VC-FB | MC-FB |
|---|---|---|---|---|---|
| RND | all domains | all tasks | 389 | 390 | 396 |
| DIAYN | all domains | all tasks | 269 | 280 | 283 |
| RANDOM | all domains | all tasks | 111 | 131 | 133 |
| ALL | all domains | all tasks | 256 | 267 | **271** |

outperform all methods and reach 135% of CQL's score. On the RANDOM dataset, all methods perform similarly poorly, except for CQL on Jaco, which outperforms all methods. However, in general, these results suggest the RANDOM dataset is not informative enough to extract valuable policies–discussed further in response to Q3. There appears to be little correlation between the type of domain (Appendix A.1) and the score achieved by any method.

**Q3.** We report the aggregated performance of all FB methods across domains when trained on the full datasets in Table 1–a full breakdown of results in provided in Appendix D. Both conservative FB variants slightly exceed the performance of vanilla FB in expectation. The largest relative performance improvement is on the RANDOM dataset–MC-FB performance is 20% higher than FB, compared to 5% higher on DIAYN and 2% higher on RND. This corroborates the hypothesis that RANDOM-100K was not informative enough to extract valuable policies. These results suggest we can safely adopt conservatism into FB without worrying about performance trade-offs.

## 5 DISCUSSION AND LIMITATIONS

**Performance discrepancy between conservative variants.** Why does VC-FB outperform MC-FB on the 100k datasets, but not on the full datasets? To understand, we inspect the regularising effect of both models more closely. VC-FB regularises OOD actions on $F(s, a, z)^\top z$, with $s \sim \mathcal{D}$, and $z \sim \mathcal{Z}$, whilst MC-FB regularises OOD actions on $F(s, a, z)^\top B(s_+)$, with $(s, s_+) \sim \mathcal{D}$ and $z \sim \mathcal{Z}$. Note the trailing $z$ in VC-FB is replaced with $B(s_+)$ which ties the updates of MC-FB to $\mathcal{D}$ yet further. We hypothesised that as $|\mathcal{D}|$ reduces, $B(s_+)$ provides poorer task coverage than $z \sim \mathcal{Z}$, hence the comparable performance on full datasets and divergent performance on 100k datasets.

To test this, we evaluate a third conservative variant called *directed* (D)VC-FB which replaces all $z \sim \mathcal{Z}$ in VC-FB with $B(s_+)$ such that OOD actions are regularised on $F(s, a, B(s_+))^\top B(s_+)$ with $(s, s_+) \sim \mathcal{D}$. This ties conservative updates entirely to $\mathcal{D}$, and according to our above hypothesis, $D$VC-FB should perform worse than VC-FB and MC-FB on the 100k datasets. See Appendix B.1.6 for implementation details. We evaluate this variant on all 100k datasets, domains and tasks and compare with FB, VC-FB and MC-FB in Table 2. See Appendix C for a full breakdown.

We find the aggregate relative performance of each method is as expected i.e. $D$VC-FB < MC-FB < VC-FB, and as a consequence conclude that, for small datasets with no prior knowledge of the dataset or test tasks, VC-FB should be preferred by practitioners. Of course, for a specific domain-dataset pair, $B(s_+)$ with $s_+ \sim \mathcal{D}$ may happen to cover the tasks well, and MC-FB may outperform VC-FB. We suspect this was the case for all datasets on the Jaco domain for example. Establishing whether this will be true *a priori* requires either relaxing the restrictions imposed by the zero-shot RL setting, or better understanding of the distribution of tasks in $z$-space and their relationship to pre-training datasets. The latter is important future work.

**Computational expense of conservative variants.** The max value approximator used by the conservative FB variants performs log-sum-exponentials and concatenations across large tensors, both of which are expensive operations. We find that these operations, which are the primary contributors

Table 2: **Aggregated performance of conservative variants employing differing $z$ sampling procedures.** $D$VC-FB derives all $z$s from the backward model; VC-FB derives all $z$s from $\mathcal{Z}$; and MC-FB combines both. Performance correlates with the degree to which $z \sim \mathcal{Z}$.

| Dataset | Domain | Task | FB | $D$VC-FB | MC-FB | VC-FB |
|---|---|---|---|---|---|---|
| ALL (100k) | all domains | all tasks | 99 | 108 | 136 | 148 |

to the additional run-time, increase the training duration by approximately $3\times$ over vanilla FB. An FB training run takes approximately 4 hours on an A100 GPU, whereas the conservative FB variants take approximately 12 hours. It seems highly likely that more elegant implementations exist that would improve training efficiency. We leave such an exploration for future work.

**Learning instability.** We report the learning curves for all algorithms across domains, datasets, and tasks in Appendix H. We note many instances of instability which would require practitioners to invoke early stopping. However, both CQL and offline TD3, our task-specific baselines, exhibit similar instability, so we do not consider this behaviour to be an inherent flaw of any method, but rather an indication of the difficulty of learning representations from sub-optimal data. Future work that stabilises FB learning dynamics could boost performance and simplify their deployment by negating the need for early stopping.

We provide detail of negative results in Appendix F to help inform future research.

## 6 RELATED WORK

**Conservatism in offline RL.** Offline RL algorithms require regularisation of policies, value functions, models, or a combination to manage the offline-to-online distribution shift (Levine et al., 2020). Past works regularise policies with explicit constraints (Wu et al., 2019; Fakoor et al., 2021; Fujimoto et al., 2019; Ghasemipour et al., 2021; Peng et al., 2023; Kumar et al., 2019b; Wu et al., 2022; Yang et al., 2022b), via important sampling (Precup et al., 2001; Sutton et al., 2016; Liu et al., 2019; Nachum et al., 2019; Gelada & Bellemare, 2019), by leveraging uncertainty in predictions (Wu et al., 2021; Zanette et al., 2021; Bai et al., 2022; Jin et al., 2021), or by minimising OOD action queries (Wang et al., 2018; Chen et al., 2020b; Kostrikov et al., 2021), a form of imitation learning (Schaal, 1996; 1999). Other works constrain value function approximation so OOD action values are not overestimated (Kumar et al., 2020; 2019a; Ma et al., 2021a;b; Yang et al., 2022a). Offline model-based RL methods use the model to identify OOD states and penalise predicted rollouts passing through them (Yu et al., 2020; Kidambi et al., 2020; Yu et al., 2021; Argenson & Dulac-Arnold, 2020; Matsushima et al., 2020; Rafailov et al., 2021). All of these works have focused on regularising a finite number of policies; in contrast we extend this line of work to the zero-shot RL setting which is concerned with learning an infinite family of policies.

**Zero-shot RL.** Zero-shot RL methods leverage SFs (Borsa et al., 2018) or FB representations (Touati & Ollivier, 2021; Touati et al., 2023), each generalisations of successor features (Barreto et al., 2017), successor measures (Blier et al., 2021), universal value function approximators (Schaul et al., 2015) and successor representations (Dayan, 1993). A representation learning method is required to learn the features for SFs, with past works using inverse curiosity modules (Pathak et al., 2017), diversity methods (Liu & Abbeel, 2021; Hansen et al., 2019), Laplacian eigenfunctions (Wu et al., 2018), or contrastive learning (Chen et al., 2020a). No works have yet explored the issues arising when training these methods on low quality offline datasets. A concurrent line of work treats zero-shot RL as a sequence modelling problem (Chen et al., 2021; Janner et al., 2021; Lee et al., 2022; Reed et al., 2022; Zheng et al., 2022; Chebotar et al., 2023; Furuta et al., 2021; Siebenborn et al., 2022; Yamagata et al., 2023; Xu et al., 2022), but, unlike SF and FB, these methods do not have a robust mechanism for generalising to any task at test time. We direct the reader to Yang et al. (2023) for a comprehensive review of such methods.

## 7 CONCLUSION

In this paper, we explored training agents to perform zero-shot reinforcement learning (RL) from low quality data. We established that the existing state-of-the-art method, FB representations, suffer in this regime because they overestimate the value of out-of-distribution state-action values. As a resolution, we proposed a family of *conservative* FB algorithms that suppress either the values (VC-FB) or measures (MC-FB) of out-of-distribution state-action pairs. In experiments across various domains, tasks and datasets, we showed our proposals outperform the existing state-of-the-art by up to 150% in aggregate and surpass our task-specific baseline despite lacking access to reward labels *a priori*. In addition to improving performance when trained on sub-optimal datasets, we showed that performance on large, diverse datasets does not suffer as a consequence of our design decisions. Our proposals are a step towards the use of zero-shot RL methods in the real world.

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

# APPENDICES

## A EXPERIMENTAL DETAILS

### A.1 DOMAINS

We consider two locomotion and two goal-directed domains from the ExORL benchmark (Yarats et al., 2022) which is built atop the DeepMind Control Suite (Tassa et al., 2018). Environments are visualised here: https://www.youtube.com/watch?v=rAai4QzcYbs. The domains are summarised in Table 3.

**Walker.** A two-legged robot required to perform locomotion starting from bent-kneed position. The state and action spaces are 24 and 6-dimensional respectively, consisting of joint torques, velocities and positions. ExORL provides four tasks `stand`, `walk`, `run` and `flip`. The reward function for `stand` motivates straightened legs and an upright torse; `walk` and `run` are supersets of `stand` including reward for small and large degrees of forward velocity; and `flip` motivates angular velocity of the torso after standing. Rewards are dense.

**Quadruped.** A four-legged robot required to perform locomotion inside a 3D maze. The state and action spaces are 78 and 12-dimensional respectively, consisting of joint torques, velocities and positions. ExORL provides five tasks `stand`, `roll`, `roll fast`, `jump` and `escape`. The reward function for `stand` motivates a minimum torse height and straightened legs; `roll` and `roll fast` require the robot to flip from a position on its back with varying speed; `jump` adds a term motivating vertical displacement to stand; and `escape` requires the agent to escape from a 3D maze. Rewards are dense.

**Point-mass Maze.** A 2D maze with four rooms where the task is to move a point-mass to one of the rooms. The state and action spaces are 4 and 2-dimensional respectively; the state space consists of $x, y$ positions and velocities of the mass, the action space is the $x, y$ tilt angle. ExORL provides four reaching tasks `top left`, `top right`, `bottom left` and `bottom right`. The mass is always initialised in the top left and the reward is proportional to the distance from the goal, though is sparse i.e. it only registers once the agent is reasonably close to the goal.

**Jaco.** A 3D robotic arm tasked with reaching an object. The state and action spaces are 55 and 6-dimensional respectively and consist of joint torques, velocities and positions. ExORL provides four reaching tasks `top left`, `top right`, `bottom left` and `bottom right`. The reward is proportional to the distance from the goal object, though is sparse i.e. it only registers once the agent is reasonably close to the goal object.

Table 3: **Experimental domain summary.** *Dimensionality* refers to the relative size of state and action spaces. *Type* is the task categorisation, either locomotion (satisfy a prescribed behaviour until the episode ends) or goal-reaching (achieve a specific task to terminate the episode). *Reward* is the frequency with which non-zero rewards are provided, where dense refers to non-zero rewards at every timestep and sparse refers to non-zero rewards only at positions close to the goal. Green and red colours reflect the relative difficulty of these settings.

| Domain | Dimensionality | Type | Reward |
|---|---|---|---|
| Walker | Low | Locomotion | Dense |
| Quadruped | High | Locomotion | Dense |
| Point-mass Maze | Low | Goal-reaching | Sparse |
| Jaco | High | Goal-reaching | Sparse |

### A.2 DATASETS

We train on 100,000 transitions uniformly sampled from three datasets on the ExORL benchmark collected by different unsupervised agents: RANDOM, DIAYN, and RND. The state coverage on Point-mass maze is depicted in Figure 6. Though harder to visualise, we found that state marginals on higher-dimensional tasks (e.g. Walker) showed a similar diversity in state coverage.

**RND.** An agent whose exploration is directed by the predicted error in its ensemble of dynamics models. Informally, we say RND datasets exhibit *high* state diversity.

**DIAYN**. An agent that attempts to sequentially learn a set of skills. Informally, we say DIAYN datasets exhibit *medium* state diversity.

**RANDOM**. An agent that selects actions uniformly at random from the action space. Informally, we say RANDOM datasets exhibit *low* state diversity.

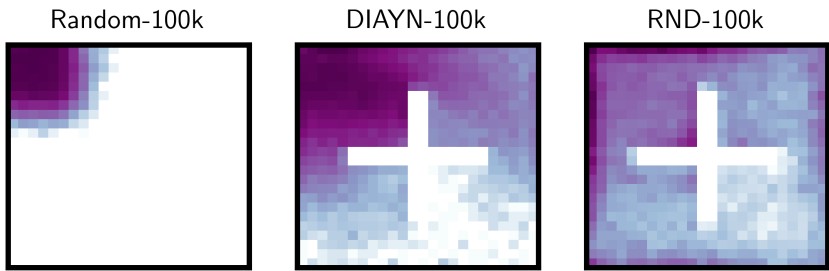

| Random-100k | DIAYN-100k | RND-100k |
| --- | --- | --- |

Figure 6: **Point-mass maze state coverage by dataset.** *(left)* RANDOM; *(middle)* DIAYN; *(right)* RND.

## B  IMPLEMENTATIONS

Here we detail implementations for all methods discussed in this paper. The code required to reproduce our experiments is provided open-source at: `https://anonymous.4open.science/r/conservative-world-models-4903`.

### B.1  FORWARD-BACKWARD REPRESENTATIONS

#### B.1.1  ARCHITECTURE

The forward-backward architecture described below follows the implementation by Touati et al. (2023) exactly, other than the batch size which we reduce from 1024 to 512. We did this to reduce the computational expense of each run without limiting performance. The hyperparameter study in Appendix J of Touati et al. (2023) shows this choice is unlikely to affect FB performance. All other hyperparameters are reported in Table 4.

**Forward Representation** $F(s, a, z)$. The input to the forward representation $F$ is always preprocessed. State-action pairs $(s, a)$ and state-task pairs $(s, z)$ have their own preprocessors $P_F^1$ and $P_F^2$. $P_F^1$ and $P_F^2$ are feedforward MLPs that embed their inputs into a 512-dimensional space. These embeddings are concatenated and passed through a third feedforward MLP $F$ which outputs a $d$-dimensional embedding vector.

**Backward Representation** $B(s)$. The backward representation $B$ is a feedforward MLP that takes a state as input and outputs a $d$-dimensional embedding vector.

**Actor** $\pi(s, z)$. Like the forward representation, the inputs to the policy network are similarly preprocessed. State-action pairs $(s, a)$ and state-task pairs $(s, z)$ have their own preprocessors $P_\pi^1$ and $P_\pi^2$. $P_\pi^1$ and $P_\pi^2$ are feedforward MLPs that embed their inputs into a 512-dimensional space. These embeddings are concatenated and passed through a third feedforward MLP which outputs a $a$-dimensional vector, where $a$ is the action-space dimensionality. A Tanh activation is used on the last layer to normalise their scale. As per Fujimoto et al. (2019)'s recommendations, the policy is smoothed by adding Gaussian noise $\sigma$ to the actions during training.

**Misc.** Layer normalisation (Ba et al., 2016) and Tanh activations are used in the first layer of all MLPs to standardise the inputs.

#### B.1.2  $z$ SAMPLING

FB representations require a method for sampling the task vector $z$ at each learning step. Touati et al. (2023) employ a mix of two methods, which we replicate:

1. Uniform sampling of $z$ on the hypersphere surface of radius $\sqrt{d}$ around the origin of $\mathbb{R}^d$,

Table 4: **(VC/MC)** -FB Hyperparameters. The additional hyperparameters for Conservative FB representations are highlighted in blue .

| Hyperparameter | Value |
|---|---|
| Latent dimension $d$ | 50 (100 for maze) |
| $F$ hidden layers | 2 |
| $F$ hidden dimension | 1024 |
| $B$ hidden layers | 3 |
| $B$ hidden dimension | 256 |
| $P_F$ hidden layers | 2 |
| $P_F$ hidden dimension | 1024 |
| $P_\pi$ hidden layers | 2 |
| $P_\pi$ hidden dimension | 1024 |
| Std. deviation for policy smoothing $\sigma$ | 0.2 |
| Truncation level for policy smoothing | 0.3 |
| Learning steps | 1,000,000 |
| Batch size | 512 |
| Optimiser | Adam |
| Learning rate | 0.0001 |
| Discount $\gamma$ | 0.98 (0.99 for maze) |
| Activations (unless otherwise stated) | ReLU |
| Target network Polyak smoothing coefficient | 0.01 |
| $z$-inference labels | 100,000 |
| $z$ mixing ratio | 0.5 |
| Conservative budget $\tau$ | 50 |
| OOD action samples per policy $N$ | 3 |

2. Biased sampling of $z$ by passing states $s \sim \mathcal{D}$ through the backward representation $z = B(s)$. This also yields vectors on the hypersphere surface due to the $L2$ normalisation described above, but the distribution is non-uniform.

We sample $z$ 50:50 from these methods at each learning step.

### B.1.3   MAXIMUM VALUE APPROXIMATOR $\mu$

The conservative variants of FB require access to a policy distribution $\mu(a|s)$ that maximises the value of the current $Q$ iterate in expectation. Recall the standard CQL loss

$$\mathcal{L}_{\text{CQL}} = \alpha \cdot \left( \mathbb{E}_{s\sim\mathcal{D}, a\sim\mu(a|s)}[Q(s,a)] - \mathbb{E}_{(s,a)\sim\mathcal{D}}[Q(s,a)] - R(\mu) \right) + \mathcal{L}_Q, \tag{10}$$

where $\alpha$ is a scaling parameter, $\mu(a|s)$ the policy distribution we seek, $R$ regularises $\mu$ and $\mathcal{L}_Q$ represents the normal TD loss on $Q$. Kumar et al. (2020)'s most performant CQL variant (CQL($\mathcal{H}$)) utilises maximum entropy regularisation on $\mu$ i.e. $R = \mathcal{H}(\mu)$. They show that obtaining $\mu$ can be cast as a closed-form optimisation problem of the form:

$$\max_\mu \mathbb{E}_{x\sim\mu(x)}[f(x)] + \mathcal{H}(\mu) \text{ s.t.} \sum_x \mu(x) = 1, \mu(x) \geq 0 \,\forall x, \tag{11}$$

and has optimal solution $\mu^*(x) = \frac{1}{Z}\exp(f(x))$, where $Z$ is a normalising factor. Plugging Equation 11 into Equation 10 we obtain:

$$\mathcal{L}_{\text{CQL}} = \alpha \cdot \left( \mathbb{E}_{s\sim\mathcal{D}}[\log \sum_a \exp(Q(s,a))] - \mathbb{E}_{(s,a)\sim\mathcal{D}}[Q(s,a)] \right) + \mathcal{L}_Q. \tag{12}$$

In discrete action spaces the `logsumexp` can be computed exactly; in continuous action spaces Kumar et al. (2020) approximate it via importance sampling using actions sampled uniformly at random, actions from the current policy conditioned on $s_t \sim \mathcal{D}$, and from the current policy conditioned on $s_{t+1} \sim \mathcal{D}$[4]:

---

[4]Conditioning on next states $s_{t+1} \sim \mathcal{D}$ is not mentioned in the paper, but is present in their official implementation.

$$\log \sum_a \exp Q(s_t, a_t) = \log(\frac{1}{3} \sum_a \exp Q(s_t, a_t)) + \frac{1}{3} \sum_a \exp Q(s_t, a_t)) + \frac{1}{3} \sum_a \exp(\exp Q(s_t, a_t)),$$

$$= \log(\frac{1}{3} \mathbb{E}_{a_t \sim \text{Unif}(\mathcal{A})} \left[ \frac{\exp(Q(s_t, a_t)}{\text{Unif}(\mathcal{A})} \right] + \frac{1}{3} \mathbb{E}_{a_t \sim \pi(a_t|s_t)} \left[ \frac{\exp(Q(s_t, a_t))}{\pi(a_t|s_t)} \right]$$

$$\frac{1}{3} \mathbb{E}_{a_{t+1} \sim \pi(a_{t+1}|s_{t+1})} \left[ \frac{\exp(Q(s_t, a_t))}{\pi(a_{t+1}|s_{t+1})} \right]),$$

$$= \log(\frac{1}{3N} \sum_{a_i \sim \text{Unif}(\mathcal{A})}^{N} \left[ \frac{\exp(Q(s_t, a_t))}{\text{Unif}(\mathcal{A})} \right] + \frac{1}{6N} \sum_{a_i \sim \pi(a_t|s_t)}^{2N} \left[ \frac{\exp(Q(s_t, a_t))}{\pi(a_i|s_t)} \right]$$

$$\frac{1}{3N} \sum_{a_i \sim \pi(a_{t+1}|s_{t+1})}^{N} \left[ \frac{\exp(Q(s_t, a_t))}{\pi(a_i|s_{t+1})} \right]),$$

$$(13)$$

with $N$ a hyperparameter defining the number of actions to sample across the action-space. We can substitute $F(s, a, z)^\top z$ for $Q(s, a)$ in the final expression of Equation 14 to obtain the equivalent for VC-FB:

$$\log \sum_a \exp F(s_t, a_i, z)^\top z = \log(\frac{1}{3N} \sum_{a_i \sim \text{Unif}(\mathcal{A})}^{N} \left[ \frac{\exp(F(s_t, a_i, z)^\top z)}{\text{Unif}(\mathcal{A})} \right] + \frac{1}{6N} \sum_{a_i \sim \pi(a_t|s_t)}^{2N} \left[ \frac{\exp(F(s_t, a_i, z^\top z)}{\pi(a_i|s_t)} \right]$$

$$\frac{1}{3N} \sum_{a_i \sim \pi(a_{t+1}|s_{t+1})}^{N} \left[ \frac{\exp(F(s_t, a_i, z)^\top z)}{\pi(a_i|s_{t+1})} \right]).$$

$$(14)$$

In Appendix G, Figure 11 we show how the performance of VC-FB varies with the number of action samples. In general, performance improves with the number of action samples, but we limit $N = 3$ to limit computational burden. The formulation for MC-FB is identical other than each value $F(s, a, z)^T z$ being replaced with measures $F(s, a, z)^T B(s_+)$.

### B.1.4 Dynamically Tuning $\alpha$

A critical hyperparameter is $\alpha$ which weights the conservative penalty with respect to other losses during each update. We initially trialled constant values of $\alpha$, but found performance to be fragile to this selection, and lacking robustness across environments. Instead, we follow Kumar et al. (2020) once again, and instantiate their algorithm for dynamically tuning $\alpha$, which they call Lagrangian dual gradient-descent on $\alpha$. We introduce a conservative budget parameterised by $\tau$, and set $\alpha$ with respect to this budget:

$$\min_{FB} \max_{\alpha \geq 0} \alpha \cdot \left( \mathbb{E}_{s \sim \mathcal{D}, a \sim \mu(a|s)z \sim \mathcal{Z}}[F(s, a, z)^\top z] - \mathbb{E}_{(s,a) \sim \mathcal{D}, z \sim \mathcal{Z}}[F(s, a, z)^\top z] - \tau \right) + \mathcal{L}_{\text{FB}}. \quad (15)$$

Intuitively, this implies that if the scale of overestimation $\leq \tau$ then $\alpha$ is set close to 0, and the conservative penalty does not affect the updates. If the scale of overestimation $\geq \tau$ then $\alpha$ is set proportionally to this gap, and thus the conservative penalty is proportional to the degree of overestimation above $\tau$. As above, for the MC-FB variant values $F(s, a, z)^\top z$ are replaced with measures $F(s, a, z)^\top B(s_+)$.

### B.1.5 Algorithm

We summarise the end-to-end implementation of VC-FB as pseudo-code in Algorithm 1. MC-FB representations are trained identically other than at line 10 where the conservative penalty is computed for $M$ instead of $Q$, and in line 12 where $M$s are lower bounded via Equation 9.

---

**Algorithm 1** Pre-training value-conservative forward-backward representations

---

**Require:** $\mathcal{D}$: dataset of trajectories
               $F_{\theta_F}, B_{\theta_B}, \pi$: randomly initialised networks
               $N, \mathcal{Z}, \nu, b$: learning steps, z-sampling distribution, polyak momentum, batch size

1: **for** learning step $n = 1...N$ **do**
2:    $\{(s_i, a_i, s_{i+1})\} \sim \mathcal{D}_{i \in |b|}$                        ◁ *Sample mini-batch of transitions*
3:    $\{z_i\}_{i \in |b|} \sim \mathcal{Z}$                              ◁ *Sample zs (Appendix B.1.2)*
4:
5:    *// FB Update*
6:    $\{a_{i+1}\} \sim \pi(s_{i+1}, z_i)$             ◁ *Sample batch of actions at next states from policy*
7:    Update $FB$ given $\{(s_i, a_i, s_{i+1}, a_{i+1}, z_i)\}$                 ◁ *Equation 5*
8:
9:    *// Conservative Update*
10:   $Q^{\max}(s_i, a_i) \approx \log \sum_a \exp F(s_i, a_i, z_i)^\top z_i$  ◁ *Compute conservative penalty (Equation 14)*

11:   Compute $\alpha$ given $Q^{\max}$ via Lagrangian dual gradient-descent        ◁ *Equation 15*
12:   Lower bound $Q$                                          ◁ *Equation* **??**
13:
14:   *// Actor Update*
15:   $\{a_i\} \sim \pi(s_i, z_i)$                               ◁ *Sample actions from policy*
16:   Update actor to maximise $\mathbb{E}[F(s_i, a_i, z_i)^\top z_i]$      ◁ *Standard actor-critic formulation*

17:
18:   *// Update target networks via polyak averaging*
19:   $\theta_F^- \leftarrow \nu\theta_F^- + (1-\nu)\theta_F$                      ◁ *Forward target network*
20:   $\theta_B^- \leftarrow \nu\theta_B^- + (1-\nu)\theta_B$                     ◁ *Backward target network*
21: **end for**

---

### B.1.6  *Directed* VALUE-CONSERVATIVE FORWARD BACKWARD REPRESENTATIONS

VC-FB applies conservative updates using task vectors $z$ sampled from $\mathcal{Z}$ (which in practice is a uniform distribution over the $\sqrt{d}$-hypersphere). This will include many vectors corresponding to tasks that are never evaluated in practice in downstream applications. Intuitively, it may seem reasonable to direct conservative updates to focus on tasks that are likely to be encountered downstream. One simple way of doing this would be consider the set of all goal-reaching tasks for goal states in the training distribution, which corresponds to sampling $z = B(s_g)$ for some $s_g \sim \mathcal{D}$. This leads to the following conservative loss function:

$$\mathcal{L}_{\text{DVC-FB}} = \alpha \cdot \Big( \mathbb{E}_{s \sim \mathcal{D}, a \sim \mu(a|s), s_g \sim \mathcal{D}} [F(s, a, B(s_g))^\top B(s_g)]$$

$$- \mathbb{E}_{(s,a) \sim \mathcal{D}, s_g \sim \mathcal{D}} [F(s, a, B(s_g))^\top B(s_g)] - \mathcal{H}(\mu) \Big) + \mathcal{L}_{\text{FB}}. \quad (16)$$

We call models learnt via this loss *directed*-VC-FB (*D*VC-FB). While we were initially open to the possibility that *D*VC-FB updates would be better targeted than those of VC-FB, and would lead to improved downstream task performance, this turns out not to be the case in our experimental settings. This may be because our training datasets $\mathcal{D}$ are so heavily concentrated in state space that conservatism ends up being applied to a very narrow region of $z$-space that is even less representative of the downstream evaluation tasks than undirected sampling $z \sim \mathcal{Z}$. This would be less of an issue when training on larger, more exploratory datasets, but these are also the datasets on which conservatism is less likely to be necessary in the first place. We report scores obtained by the *D*VC-FB method across all 100k datasets, domains and tasks in Appendix C.

### B.2  SUCCESSOR FEATURES

We directly reimplement SFs, with basic features $\varphi(s)$ provided by Laplacian eigenfunctions (Wu et al., 2018), from Touati et al. (2023).

Table 5: **SF Hyperparameters.**

| Hyperparameter | Value |
| --- | --- |
| Latent dimension $d$ | 50 (100 for maze) |
| $\psi$ hidden layers | 2 |
| $\psi$ hidden dimension | 1024 |
| $\varphi$ hidden layers | 3 |
| $\varphi$ hidden dimension | 256 |
| $P_\psi$ hidden layers | 2 |
| $P_\psi$ hidden dimension | 1024 |
| $P_\pi$ hidden layers | 2 |
| $P_\pi$ hidden dimension | 1024 |
| Std. deviation for policy smoothing $\sigma$ | 0.2 |
| Truncation level for policy smoothing | 0.3 |
| Learning steps | 1,000,000 |
| Batch size | 512 |
| Optimiser | Adam |
| Learning rate | 0.0001 |
| Discount $\gamma$ | 0.98 (0.99 for maze) |
| Activations (unless otherwise stated) | ReLU |
| Target network Polyak smoothing coefficient | 0.01 |
| $z$-inference labels | 100,000 |
| $z$ mixing ratio | 0.5 |
| Regularisation weight $\lambda$ | 1 |

### B.2.1 ARCHITECTURE

**SF $\psi(s, a, z)$.** The input to the SF $\psi$ is always preprocessed. State-action pairs $(s, a)$ and state-task pairs $(s, z)$ have their own preprocessors $P_\psi^1$ and $P_\psi^2$. $P_\psi^1$ and $P_\psi^2$ are feedforward MLPs that embed their inputs into a 512-dimensional space. These embeddings are concatenated and passed through a third feedforward MLP $\psi$ which outputs a $d$-dimensional embedding vector. Note this is identical to the implementation of $F$ as described in Appendix B.1. All other hyperparameters are reported in Table 5.

**Feature Embedding $\varphi(s)$.** The feature map $\varphi(s)$ is a feedforward MLP that takes a state as input and outputs a $d$-dimensional embedding vector. The loss function for learning the feature embedding is provided in Appendix B.2.2.

**Actor $\pi(s, z)$.** Like the forward representation, the inputs to the policy network are similarly preprocessed. State-action pairs $(s, a)$ and state-task pairs $(s, z)$ have their own preprocessors $P_\pi^1$ and $P_\pi^2$. $P_\pi^1$ and $P_\pi^2$ are feedforward MLPs that embed their inputs into a 512-dimensional space. These embeddings are concatenated and passed through a third feedforward MLP which outputs a $a$-dimensional vector, where $a$ is the action-space dimensionality. A Tanh activation is used on the last layer to normalise their scale. As per Fujimoto et al. (2019)'s recommendations, the policy is smoothed by adding Gaussian noise $\sigma$ to the actions during training. Note this is identical to the implementation of $\pi(s, z)$ as described in Appendix B.1.

**Misc.** Layer normalisation (Ba et al., 2016) and Tanh activations are used in the first layer of all MLPs to standardise the inputs. $z$ sampling distribution $\mathcal{Z}$ is identical to FB's (Appendix B.1.2).

### B.2.2 LAPLACIAN EIGENFUNCTIONS LOSS

Laplacian eigenfunction features $\varphi(s)$ are learned as per Wu et al. (2018). They consider the symmetrized MDP graph Laplacian created by some policy $\pi$, defined as $L = \text{Id} - \frac{1}{2}(\mathcal{P}_\pi \text{diag}\rho^- 1 + \text{diag}\rho^- 1(\mathcal{P}_\pi)^T)$. They learn the eigenfunctions of $L$ with the following:

$$\min_\varphi \mathbb{E}_{(s_t, s_{t+1}) \sim \mathcal{D}} \left[ ||\varphi(s_t) - \varphi(s_{t+1})||^2 \right] + \lambda \mathbb{E}_{(s, s_+) \sim \mathcal{D}} \left[ (\varphi(s)^\top \varphi(s_+))^2 - ||\varphi(s)||_2^2 - ||\varphi(s_+)||_2^2 \right],$$
(17)

which comes from Koren (2003).

## B.3   CQL

### B.3.1   ARCHITECTURE

We adopt the same implementation and hyperparameters as is used on the ExORL benchmark. CQL inherits all functionality from a base soft actor-critic agent (Haarnoja et al., 2018), but adds a conservative penalty to the critic updates (Equation 7). Hyperparameters are reported in Table 6.

**Critic(s).** CQL employs double Q networks, where the target network is updated with Polyak averaging via a momentum coefficient. The critics are feedforward MLPs that take a state-action pair $(s, a)$ as input and output a value $\in \mathbb{R}^1$.

**Actor.** The actor is a standard feedforward MLP taking the state $s$ as input and outputting an $2a$-dimensional vector, where $a$ is the action-space dimensionality. The actor predicts the mean and standard deviation of a Gaussian distribution for each action dimension; during training a value is sampled at random, during evaluation the mean is used.

## B.4   TD3

### B.4.1   ARCHITECTURE

We adopt the same implementation and hyperparameters as is used on the ExORL benchmark. Hyperparameters are reported in Table 6.

**Critic(s).** TD3 employs double Q networks, where the target network is updated with Polyak averaging via a momentum coefficient. The critics are feedforward MLPs that take a state-action pair $(s, a)$ as input and output a value $\in \mathbb{R}^1$.

**Actor.** The actor is a standard feedforward MLP taking the state $s$ as input and outputting an $a$-dimensional vector, where $a$ is the action-space dimensionality. The policy is smoothed by adding Gaussian noise $\sigma$ to the actions during training.

**Misc.** As is usual with TD3, layer normalisation (Ba et al., 2016) is applied to the inputs of all networks.

Table 6: **Offline RL baseline algorithms hyperparameters.**

| Hyperparameter | CQL | TD3 |
|---|---|---|
| Critic hidden layers | 2 | 2 |
| Critic hidden dimension | 1024 | 1024 |
| Actor hidden layers | 2 | 2 |
| Actor hidden dimension | 1024 | 1024 |
| Learning steps | 1,000,000 | 1,000,000 |
| Batch size | 1024 | 1024 |
| Optimiser | Adam | Adam |
| Learning rate | 0.0001 | 0.0001 |
| Discount $\gamma$ | 0.98 (0.99 for maze) | 0.98 (0.99 for maze) |
| Activations | ReLU | ReLU |
| Target network Polyak smoothing coefficient | 0.01 | 0.01 |
| Sampled Actions Number | 3 | - |
| $\alpha$ | 0.01 | - |
| Lagrange | False | - |
| Std. deviation for policy smoothing $\sigma$ | - | 0.2 |
| Truncation level for policy smoothing | - | 0.3 |

## C 100K DATASET RESULTS

Table 7: **100k dataset experimental results.** For each dataset-domain pair, we report the score at the step for which the all-task IQM is maximised when averaging across 5 seeds, and the constituent task scores at that step. Bracketed numbers represent the 95% confidence interval obtained by a stratified bootstrap.

| Dataset | Domain | Task | TD3 | CQL | SF-LAP | FB | VC-FB | *DVC-FB* | MC-FB |
|---|---|---|---|---|---|---|---|---|---|
| | | walk | 210 (205–231) | 138 (128–140) | 58 (31–103) | 184 (108–274) | 446 (435–460) | 394 (275–512) | 247 (137–318) |
| | | stand | 362 (335–379) | 386 (374–391) | 190 (128–251) | 558 (500–637) | 624 (603–639) | 590 (557–622) | 480 (401–517) |
| | walker | run | 84 (78–90) | 71 (63–75) | 34 (27–43) | 101 (88–144) | 179 (159–194) | 134 (77–191) | 106 (72–145) |
| | | flip | 162 (148–171) | 153 (130–172) | 70 (56–84) | 163 (90–203) | 325 (294–350) | 250 (215–286) | 164 (120–198) |
| | | all tasks | 189 (177–200) | 142 (135–149) | 91 (70–107) | 266 (233–283) | 396 (381–407) | 342 (284–400) | 252 (188–288) |
| | | stand | 119 (9–342) | 167 (73–266) | 108 (51–192) | 134 (91–188) | 331 (199–405) | 269 (152–385) | 171 (71–369) |
| | | roll fast | 63 (4–180) | 93 (18–219) | 80 (22–181) | 83 (57–127) | 141 (87–191) | 146 (85–207) | 81 (21–194) |
| | quadruped | roll | 96 (8–272) | 251 (126–330) | 100 (23–305) | 139 (68–234) | 141 (98–212) | 209 (123–295) | 132 (40–251) |
| | | jump | 85 (7–255) | 128 (65–223) | 94 (28–223) | 121 (79–193) | 159 (105–212) | 167 (100–234) | 97 (37–191) |
| | | escape | 3 (0–10) | 3 (2–4) | 1 (1–4) | 7 (3–12) | 8 (3–15) | 13 (6–19) | 5 (1–12) |
| | | all tasks | 81 (6–230) | 129 (70–207) | 89 (30–170) | 93 (69–137) | 168 (104–201) | 161 (96–225) | 104 (38–212) |
| RND-100k | | reach top right | 457 (0–733) | 433 (275–558) | 1 (0–185) | 0 (0–26) | 0 (0–203) | 0 (0–0) | 99 (9–377) |
| | | reach top left | 921 (895–938) | 561 (493–717) | 302 (27–825) | 384 (0–724) | 662 (218–903) | 244 (10–477) | 723 (363–895) |
| | point-mass maze | reach bottom right | 0 (0–0) | 0 (0–0) | 0 (0–0) | 0 (0–0) | 0 (0–0) | 0 (0–0) | 0 (0–0) |
| | | reach bottom left | 85 (22–295) | 253 (102–451) | 0 (0–17) | 0 (0–0) | 479 (70–748) | 250 (0–501) | 384 (0–776) |
| | | all tasks | 345 (171–405) | 299 (262–364) | 126 (12–206) | 102 (0–181) | 323 (177–412) | 123 (59–188) | 270 (154–459) |
| | | reach top right | 0 (0–0) | 37 (21–53) | 0 (0–2) | 0 (0–3) | 1 (0–4) | 6 (3–9) | 17 (8–29) |
| | | reach top left | 0 (0–0) | 21 (12–35) | 2 (0–5) | 2 (1–4) | 2 (0–3) | 11 (7–16) | 9 (1–21) |
| | jaco | reach bottom right | 0 (0–0) | 37 (21–53) | 0 (0–0) | 0 (0–6) | 5 (2–21) | 7 (3–11) | 16 (6–25) |
| | | reach bottom left | 0 (0–0) | 20 (17–28) | 1 (0–4) | 7 (3–15) | 4 (1–21) | 3 (1–5) | 11 (1–45) |
| | | all tasks | 0 (0–0) | 31 (25–36) | 1 (0–2) | 4 (1–6) | 7 (3–12) | 7 (4–9) | 17 (7–26) |
| | **all domains** | **all tasks** | 135 | 136 | 90 | 97 | 245 | 142 | 178 |
| | | walk | 150 (132–167) | 147 (118–201) | 251 (158–315) | 93 (58–113) | 262 (141–370) | 248 (243–253) | 261 (175–351) |
| | | stand | 263 (235–306) | 406 (365–455) | 276 (189–292) | 498 (381–652) | 455 (401–492) | 387 (352–423) | 423 (375–595) |
| | walker | run | 46 (44–48) | 38 (33–43) | 53 (32–59) | 98 (79–114) | 83 (75–94) | 87 (82–92) | 81 (71–108) |
| | | flip | 163 (152–174) | 149 (116–182) | 144 (89–162) | 193 (136–212) | 229 (195–249) | 180 (155–205) | 183 (150–239) |
| | | all tasks | 154 (142–176) | 147 (134–172) | 143 (92–156) | 274 (214–301) | 252 (195–291) | 226 (208–243) | 265 (195–322) |
| | | stand | 849 (737–893) | 299 (139–405) | 313 (134–562) | 459 (397–530) | 430 (394–482) | 447 (413–482) | 458 (396–513) |
| | | roll fast | 447 (358–500) | 164 (75–195) | 185 (152–266) | 288 (256–328) | 260 (236–282) | 290 (285–296) | 293 (276–299) |
| | quadruped | roll | 709 (619–800) | 264 (128–369) | 189 (85–355) | 460 (409–492) | 415 (392–439) | 429 (407–452) | 456 (407–494) |
| | | jump | 410 (368–518) | 196 (97–280) | 240 (102–362) | 363 (318–419) | 358 (324–400) | 391 (371–411) | 373 (341–403) |
| | | escape | 23 (15–31) | 6 (3–10) | 16 (5–28) | 45 (35–58) | 32 (27–43) | 45 (42–48) | 42 (37–50) |
| | | all tasks | 487 (440–528) | 208 (98–282) | 189 (124–274) | 322 (285–364) | 296 (272–327) | 321 (307–335) | 331 (302–342) |
| DIAYN-100k | | reach top right | 796 (655–800) | 760 (489–784) | 0 (0–0) | 0 (0–0) | 0 (0–0) | 0 (0–0) | 27 (0–86) |
| | | reach top left | 943 (942–946) | 943 (941–949) | 764 (331–934) | 576 (76–876) | 911 (615–927) | 557 (270–844) | 853 (572–932) |
| | point-mass maze | reach bottom right | 0 (0–0) | 0 (0–0) | 0 (0–0) | 0 (0–0) | 0 (0–0) | 0 (0–0) | 0 (0–0) |
| | | reach bottom left | 799 (538–808) | 0 (0–0) | 0 (0–0) | 0 (0–1) | 0 (0–0) | 0 (0–0) | 0 (0–0) |
| | | all tasks | 798 (598–803) | 380 (244–392) | 190 (82–233) | 144 (19–219) | 227 (153–231) | 138 (67–210) | 213 (153–241) |
| | | reach top right | 0 (0–0) | 17 (10–31) | 8 (1–19) | 2 (0–9) | 6 (2–11) | 0 (0–0) | 9 (5–17) |
| | | reach top left | 0 (0–0) | 10 (4–18) | 0 (0–2) | 2 (0–5) | 7 (0–14) | 27 (2–53) | 0 (0–0) |
| | jaco | reach bottom right | 0 (0–0) | 17 (10–31) | 3 (2–7) | 4 (2–14) | 6 (2–14) | 0 (0–0) | 12 (2–40) |
| | | reach bottom left | 0 (0–0) | 2 (0–13) | 2 (0–3) | 10 (5–20) | 5 (1–9) | 15 (0–30) | 10 (5–18) |
| | | all tasks | 0 (0–0) | 15 (9–21) | 4 (2–7) | 8 (5–9) | 8 (5–10) | 10 (4–17) | 11 (4–16) |
| | **all domains** | **all tasks** | 320 | 177 | 166 | 209 | 239 | 182 | 239 |
| | | walk | 132 (105–156) | 126 (113–140) | 129 (112–139) | 76 (50–121) | 123 (84–140) | 38 (32–43) | 119 (59–211) |
| | | stand | 295 (251–328) | 246 (194–287) | 206 (161–263) | 238 (201–279) | 223 (206–244) | 223 (201–246) | 210 (187–239) |
| | walker | run | 58 (39–65) | 31 (23–49) | 49 (36–58) | 38 (32–48) | 40 (37–46) | 31 (25–36) | 32 (27–38) |
| | | flip | 72 (45–88) | 115 (97–126) | 100 (79–122) | 47 (40–60) | 63 (41–99) | 47 (43–52) | 44 (38–55) |
| | | all tasks | 105 (88–111) | 119 (108–131) | 120 (101–143) | 102 (91–119) | 113 (98–125) | 85 (79–91) | 108 (78–127) |
| | | stand | 264 (46–532) | 186 (125–296) | 285 (131–432) | 278 (154–493) | 269 (48–618) | 196 (108–284) | 172 (68–284) |
| | | roll fast | 151 (32–283) | 161 (70–223) | 64 (22–129) | 96 (17–195) | 43 (17–132) | 155 (89–220) | 78 (43–129) |
| | quadruped | roll | 260 (41–449) | 326 (215–434) | 111 (29–166) | 105 (53–188) | 130 (74–185) | 183 (120–246) | 178 (101–402) |
| | | jump | 189 (31–359) | 213 (93–294) | 128 (14–273) | 75 (30–155) | 78 (23–226) | 94 (67–121) | 147 (44–261) |
| | | escape | 4 (1–9) | 6 (2–9) | 2 (0–5) | 5 (2–9) | 2 (1–11) | 3 (2–5) | 6 (1–14) |
| | | all tasks | 191 (33–361) | 187 (96–271) | 125 (84–169) | 149 (93–159) | 125 (49–218) | 126 (88–164) | 126 (106–165) |
| RANDOM-100k | | reach top right | 0 (0–0) | 0 (0–0) | 0 (0–0) | 0 (0–0) | 0 (0–0) | 0 (0–0) | 0 (0–0) |
| | | reach top left | 1 (0–3) | 0 (0–0) | 3 (0–6) | 18 (0–55) | 26 (5–106) | 52 (0–104) | 10 (0–33) |
| | point-mass maze | reach bottom right | 0 (0–0) | 0 (0–0) | 0 (0–0) | 0 (0–0) | 0 (0–0) | 0 (0–0) | 0 (0–0) |
| | | reach bottom left | 0 (0–4) | 0 (0–0) | 0 (0–0) | 0 (0–0) | 0 (0–0) | 0 (0–0) | 0 (0–0) |
| | | all tasks | 0 (0–0) | 0 (0–0) | 0 (0–1) | 4 (0–13) | 6 (1–26) | 13 (0–26) | 2 (0–8) |
| | | reach top right | 34 (15–78) | 53 (45–60) | 0 (0–0) | 4 (0–19) | 0 (0–8) | 0 (0–0) | 4 (0–13) |
| | | reach top left | 3 (1–6) | 52 (24–88) | 2 (0–6) | 0 (0–0) | 13 (7–28) | 26 (10–42) | 23 (9–53) |
| | jaco | reach bottom right | 34 (15–78) | 53 (45–60) | 0 (0–4) | 0 (0–0) | 1 (1–1) | 30 (0–59) | 1 (0–6) |
| | | reach bottom left | 3 (1–4) | 32 (19–41) | 0 (0–0) | 2 (1–12) | 0 (0–0) | 16 (0–33) | 0 (0–9) |
| | | all tasks | 20 (10–42) | 45 (39–58) | 0 (0–2) | 4 (0–9) | 5 (2–10) | 18 (4–32) | 6 (4–18) |
| | **all domains** | **all tasks** | 62 | 82 | 60 | 53 | 59 | 51 | 56 |
| ALL | **all domains** | **all tasks** | 123 | 128 | 92 | 99 | 148 | 108 | 136 |

# D    FULL DATASET RESULTS

| Dataset | Domain | Task | FB | VC-FB | MC-FB |
|---|---|---|---|---|---|
| RND | walker | walk | 821 (758–883) | 864 (850–879) | 792 (728–857) |
| | | stand | 928 (925–930) | 878 (854–903) | 873 (812–934) |
| | | run | 281 (242–320) | 351 (328–374) | 343 (320–366) |
| | | flip | 525 (452–598) | 542 (513–571) | 598 (538–657) |
| | | all tasks | 639 (616–661) | 659 (647–670) | 651 (632–671) |
| | quadruped | stand | 957 (952–963) | 863 (777–950) | 949 (939–958) |
| | | roll fast | 574 (553–594) | 512 (471–553) | 565 (555–575) |
| | | roll | 920 (895–944) | 831 (741–921) | 890 (874–906) |
| | | jump | 736 (721–751) | 630 (570–690) | 705 (703–707) |
| | | escape | 94 (63–125) | 59 (50–68) | 66 (47–86) |
| | | all tasks | 656 (638–674) | 579 (522–635) | 635 (628–642) |
| | point-mass maze | reach top right | 0 (0–0) | 425 (153–698) | 270 (7–533) |
| | | reach top left | 612 (313–911) | 454 (138–769) | 773 (611–934) |
| | | reach bottom right | 0 (0–0) | 0 (0–0) | 0 (0–0) |
| | | reach bottom left | 268 (0–536) | 270 (2–539) | 1 (0–2) |
| | | all tasks | 219 (86–353) | 287 (117–457) | 261 (159–363) |
| | jaco | reach top right | 48 (39–56) | 24 (0–47) | 51 (23–79) |
| | | reach top left | 23 (6–40) | 14 (4–25) | 20 (7–33) |
| | | reach bottom right | 60 (55–65) | 5 (0–10) | 47 (15–79) |
| | | reach bottom left | 27 (12–42) | 88 (33–143) | 20 (9–30) |
| | | all tasks | 39 (29–50) | 33 (24–42) | 34 (18–51) |
| | **all domains** | **all tasks** | 389 | 390 | 396 |
| DIAYN | walker | walk | 459 (266–652) | 536 (305–766) | 519 (315–722) |
| | | stand | 478 (463–494) | 447 (422–472) | 517 (433–602) |
| | | run | 87 (81–93) | 84 (78–89) | 87 (65–110) |
| | | flip | 235 (151–319) | 251 (151–352) | 301 (213–388) |
| | | all tasks | 315 (247–383) | 329 (251–408) | 356 (273–439) |
| | quadruped | stand | 763 (725–801) | 785 (739–810) | 804 (756–851) |
| | | roll fast | 497 (480–514) | 491 (475–497) | 495 (491–498) |
| | | roll | 767 (726–808) | 785 (740–801) | 761 (736–786) |
| | | jump | 628 (587–669) | 620 (600–641) | 608 (594–622) |
| | | escape | 65 (62–69) | 69 (59–74) | 67 (55–79) |
| | | all tasks | 544 (517–572) | 550 (536–580) | 547 (535–559) |
| | point-mass maze | reach top right | 0 (0–0) | 0 (0–0) | 0 (0–0) |
| | | reach top left | 654 (565–742) | 928 (907–950) | 814 (725–903) |
| | | reach bottom right | 0 (0–0) | 0 (0–0) | 8 (0–16) |
| | | reach bottom left | 169 (0–337) | 7 (0–14) | 49 (0–98) |
| | | all tasks | 205 (171–240) | 233 (227–240) | 217 (180–254) |
| | jaco | reach top right | 4 (2–7) | 10 (5–15) | 4 (1–8) |
| | | reach top left | 5 (1–10) | 1 (0–2) | 2 (0–3) |
| | | reach bottom right | 9 (4–14) | 6 (3–8) | 7 (0–14) |
| | | reach bottom left | 25 (2–47) | 12 (6–18) | 25 (3–47) |
| | | all tasks | 11 (4–18) | 7 (7–7) | 10 (1–18) |
| | **all domains** | **all tasks** | 269 | 280 | 283 |
| RANDOM | walker | walk | 148 (70–225) | 170 (139–231) | 174 (142–243) |
| | | stand | 318 (281–355) | 343 (296–371) | 355 (298–393) |
| | | run | 51 (45–58) | 101 (74–121) | 100 (67–112) |
| | | flip | 57 (47–67) | 106 (49–117) | 103 (65–140) |
| | | all tasks | 143 (116–171) | 180 (129–205) | 183 (136–219) |
| | quadruped | stand | 417 (381–453) | 450 (390–492) | 431 (371–464) |
| | | roll fast | 110 (51–170) | 201 (120–251) | 215 (139–292) |
| | | roll | 231 (116–346) | 292 (255–388) | 303 (160–445) |
| | | jump | 287 (123–450) | 311 (261–354) | 340 (275–393) |
| | | escape | 10 (6–14) | 10 (5–12) | 12 (9–16) |
| | | all tasks | 211 (148–274) | 253 (180–315) | 260 (196–327) |
| | point-mass maze | reach top right | 0 (0–0) | 0 (0–0) | 0 (0–0) |
| | | reach top left | 309 (4–615) | 317 (5–629) | 307 (0–614) |
| | | reach bottom right | 0 (0–0) | 0 (0–0) | 0 (0–0) |
| | | reach bottom left | 0 (0–0) | 0 (0–0) | 0 (0–0) |
| | | all tasks | 77 (0–153) | 79 (1–157) | 76 (0–153) |
| | jaco | reach top right | 1 (1–1) | 2 (0–4) | 5 (0–10) |
| | | reach top left | 50 (0–100) | 9 (0–18) | 16 (0–31) |
| | | reach bottom right | 0 (0–0) | 15 (5–25) | 21 (0–42) |
| | | reach bottom left | 3 (1–6) | 18 (2–34) | 1 (0–3) |
| | | all tasks | 13 (0–27) | 11 (4–18) | 11 (5–16) |
| | **all domains** | **all tasks** | 111 | 131 | 133 |
| ALL | **all domains** | **all tasks** | 256 | 267 | 271 |

# E    CONSERVATIVE SUCCESSOR FEATURES

Our proposals focus on improving the machinery of FB representations[5], but we can apply similar methods to other zero-shot RL methods. In this section, we shall show that our proposals make

---

[5]We only focus on FB representations because of their superior zero-shot RL performance (Touati et al., 2023).

sense in the context of the primary alternative: *successor features* (Barreto et al., 2017; Borsa et al., 2018).

Successor features require a state-feature mapping $\varphi : \mathcal{S} \rightarrow \mathbb{R}^d$ which is usually obtained by some representation learning method (Barreto et al., 2017). SFs are the expected discounted sum of these features, starting in state $s_0$, taking action $a_0$ and following the task-dependent policy $\pi_z$ thereafter

$$\psi(s_0, a_0, z) \approx \mathbb{E}\left[\sum_{t \geq 0} \gamma^t \varphi(s_{t+1})|(s_0, a_0), \pi_z\right]. \tag{18}$$

SFs satisfy a Bellman equation (Borsa et al., 2018) and so can be trained using TD-learning on the Bellman residuals:

$$\mathcal{L}_{\text{SF}} = \mathbb{E}_{(s_t, a_t, s_{t+1}) \sim \mathcal{D}, z \sim \mathcal{Z}}\left(\psi(s_t, a_t, z)^\top z - \varphi(s_{t+1})^\top z - \gamma \bar{\psi}(s_{t+1}, \pi_z(s_{t+1}), z)^\top z\right)^2, \tag{19}$$

where $\bar{\psi}$ is a lagging target network updated via Polyak averaging, and $\mathcal{Z}$ is identical to that used for FB training (Appendix B.1.2). As with FB representations, the policy is training to maximise the $Q$-function defined by $\psi$:

$$\pi_z(s) \approx \operatorname{argmax}_a \psi(s, a, z)^\top z. \tag{20}$$

Like FB, SF training requires next action samples $a_{t+1} \sim \pi_z(s_{t+1})$ for the TD targets. We therefore expect SFs to suffer the same failure mode discussed in Section 3 (OOD state-action value over-estimation) and to benefit from the same remedial measures (value conservatism). Training *value-conservative successor features* (VC-SF) amounts to substituting the SF $Q$-function definition and loss for FB's in Equation **??**:

$$\mathcal{L}_{\text{VC-SF}} = \alpha \cdot \left(\mathbb{E}_{s \sim \mathcal{D}, a \sim \mu(a|s), z \sim \mathcal{Z}}[\psi(s, a, z)^\top z] - \mathbb{E}_{(s,a) \sim \mathcal{D}, z \sim \mathcal{Z}}[\psi(s, a, z)^\top z]\right) + \mathcal{L}_{\text{SF}}. \tag{21}$$

Both the maximum value approximator $\mu(a|s)$ (Equation 14, Section B.1.3) and $\alpha$-tuning (Equation 15, Section B.1.4) can be extracted identically to the FB case with any occurrence of $F(s, a, z)^\top z$ substituted with $\psi(s, a, z)^\top z$. As SFs do not predict successor measures we cannot formulate measure-conservative SFs.

## F   Negative Results

In this section we provide detail on experiments we attempted, but which did not provide results significant enough to be included in the main body.

### F.1   Downstream Finetuning

If we relax the zero-shot requirement, could pre-trained conservative FB representations be finetuned on new tasks or domains? Base CQL models have been finetuned effectively on unseen tasks using both online and offline data (Kumar et al., 2022), and we had hoped to replicate similar results with VC-FB and MC-FB. We ran offline and online finetuning experiments and provide details on their setups and results below. All experiments were conducted on the Walker domain.

**Offline finetuning.** We considered a setting where models are trained on a low quality dataset initially, before a high quality dataset becomes available downstream. We used models trained on the RANDOM-100k dataset and finetuned them on both the full RND and RND-100k datasets, with models trained from scratch used as our baseline. Finetuning involved the usual training protocol as described in Algorithm 1, but we limited the number of learning steps to 250k.

We found that though performance improved during finetuning, it improved no quicker than the models trained from scratch. This held for both the full RND and RND-100k datasets. We conclude

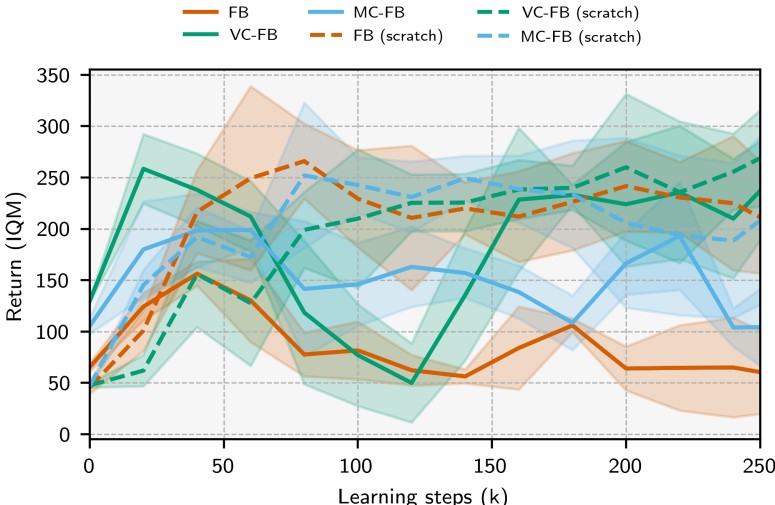

Figure 7: **Learning curves for methods finetuned on the full RND dataset.** Solid lines represent base models trained on RANDOM-100k, then finetuned; dashed lines represent models trained from scratch. The finetuned models perform no better than models trained from scratch after 250k learning steps, suggesting model re-training is currently a better strategy than offline finetuning.

that the parameter initialisation delivered after training on a low quality dataset does not obviously expedite learning when high quality data becomes available.

**Online finetuning.** We considered the online finetuning setup where a trained representation is deployed in the target environment, required to complete a specified task, and allowed to collect a replay buffer of reward-labelled online experience. We followed a standard online RL protocol where a batch of transitions was sampled from the online replay buffer after each environment step for use in updating the model's parameters. We experimented with fixing $z$ to the target task during in the actor updates (Line 16, Algorithm 1), but found it caused a quick, irrecoverable collapse in actor performance. This suggested uniform samples from $\mathcal{Z}$ provide a form of regularisation. We granted the agents 500k steps of interaction for online finetuning.

We found that performance never improved beyond the pre-trained (init) performance during fine-tuning. We speculated that this was similar to the well-documented failure mode of online finetuning of CQL (Nakamoto et al., 2023), namely taking sub-optimal actions in the real env, observing unexpectedly high reward, and updating their policy toward these sub-optimal actions. But we note that FB representations do not update w.r.t observed rewards, and so conclude this cannot be the failure mode. Instead it seems likely that FB algorithms cannot use the narrow, unexploratory experience obtained from attempting to perform a specific task to improve model performance.

We believe resolving issues associated with finetuning conservative FB algorithms once the zero-shot requirement is relaxed is an important future direction and hope that details of our negative attempts to this end help facilitate future research.

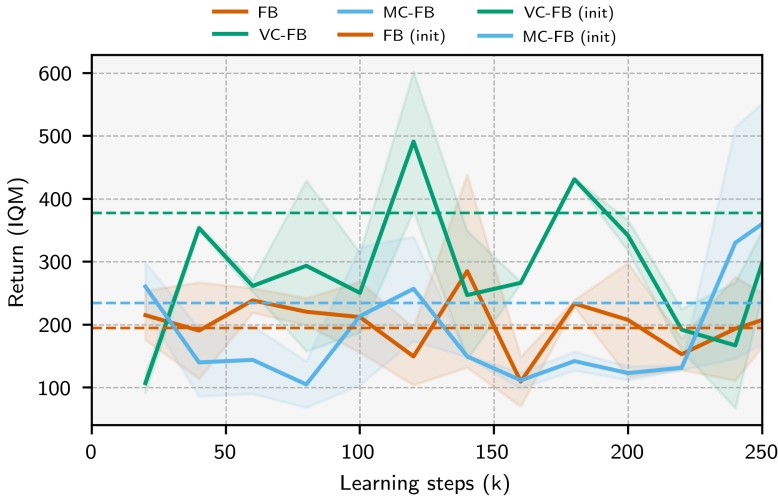

Figure 8: **Learning curves for online finetuning.** The performance at the end of pre-training (init performance) is plotted as a dashed line for each method. None of the methods consistently outperform their init performance after 250k online transitions.

## G  HYPERPARAMETER SENSITIVITY

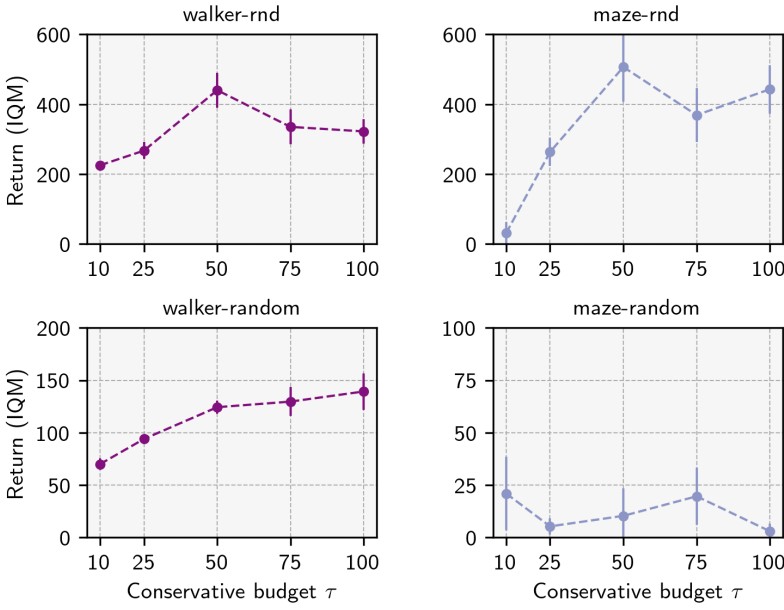

Figure 9: **VC-FB sensitivity to conservative budget $\tau$ on Walker and Point-mass Maze.** Top: RND dataset; bottom: RANDOM dataset. Maximum IQM return across the training run averaged over 3 random seeds

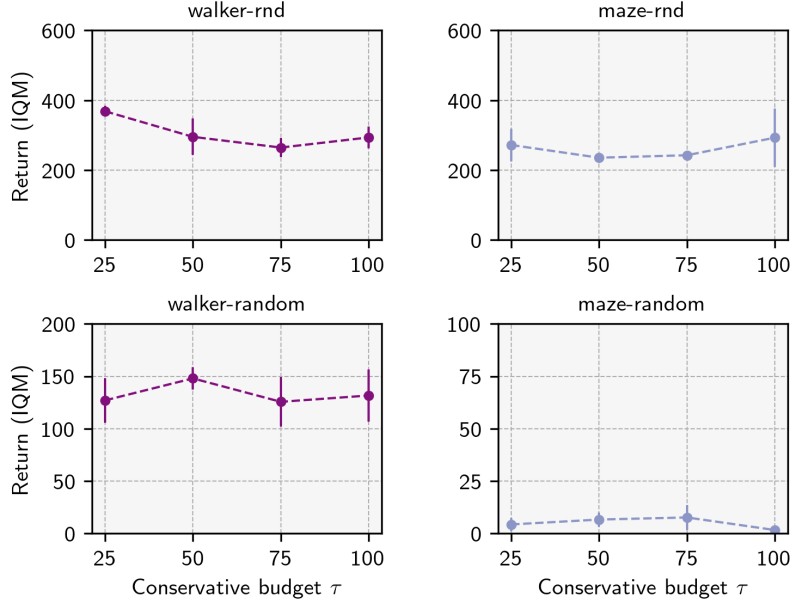

Figure 10: **MC-FB sensitivity to conservative budget $\tau$ on Walker and Point-mass Maze**. Top: RND dataset; bottom: RANDOM dataset. Maximum IQM return across the training run averaged over 3 random seeds

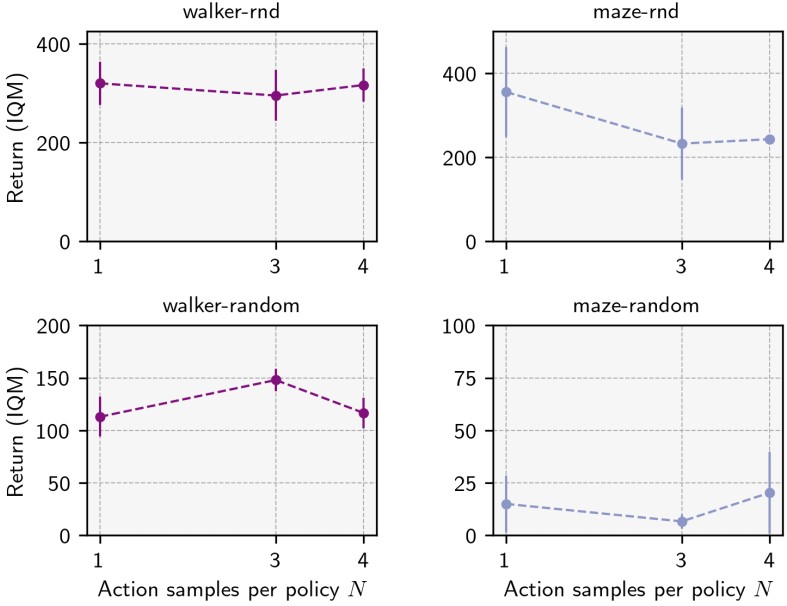

Figure 11: **MC-FB sensitivity to action samples per policy $N$ on Walker and Point-mass Maze**. Top: RND dataset; bottom: RANDOM dataset. Maximum IQM return across the training run averaged over 3 random seeds.

# H    LEARNING CURVES

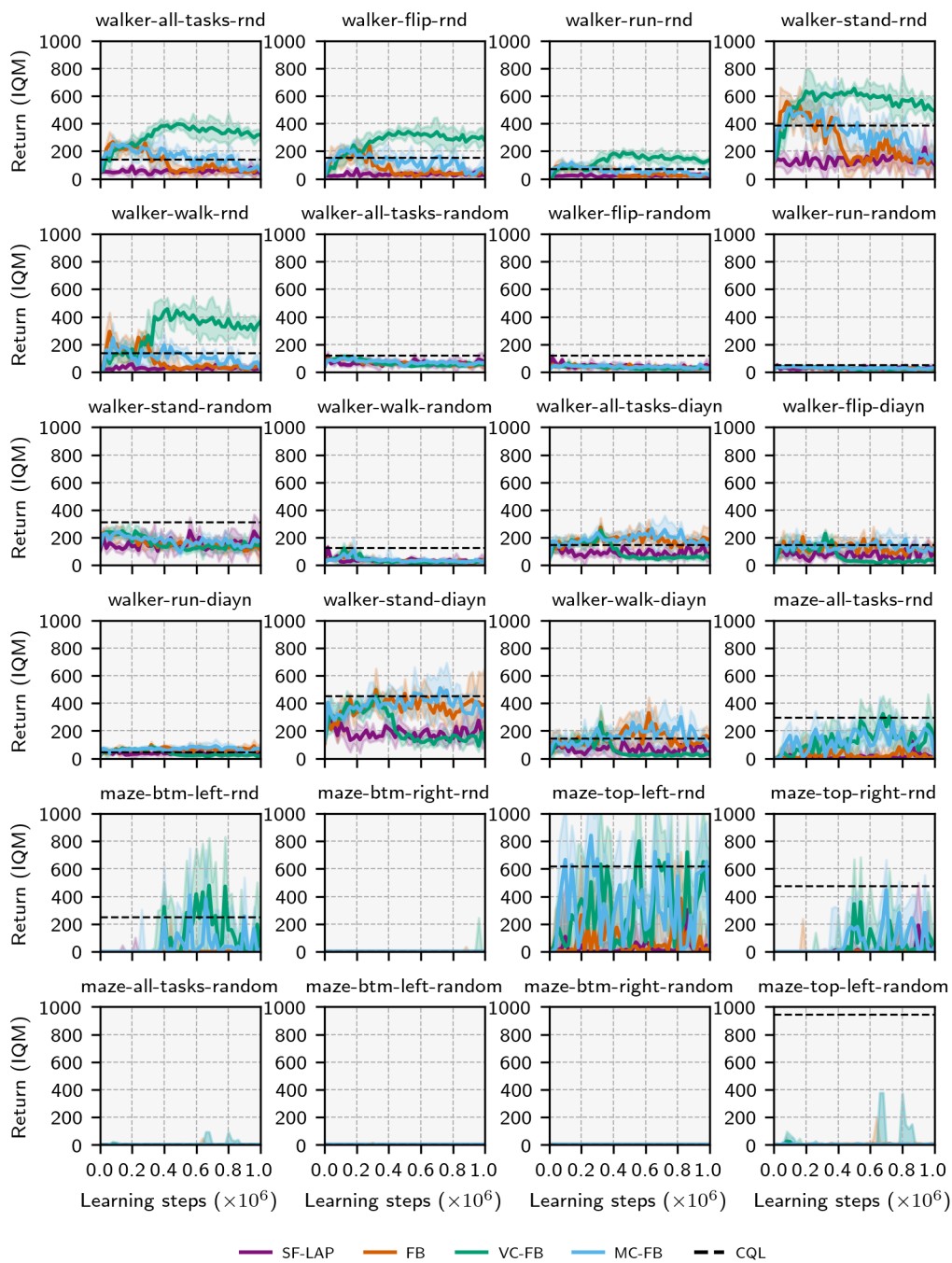

Figure 12: **Learning Curves (1/3)**. Models are evaluated every 20,000 timesteps where we perform 10 rollouts and record the IQM. Curves are the IQM of this value across 5 seeds; shaded areas are one standard deviation.

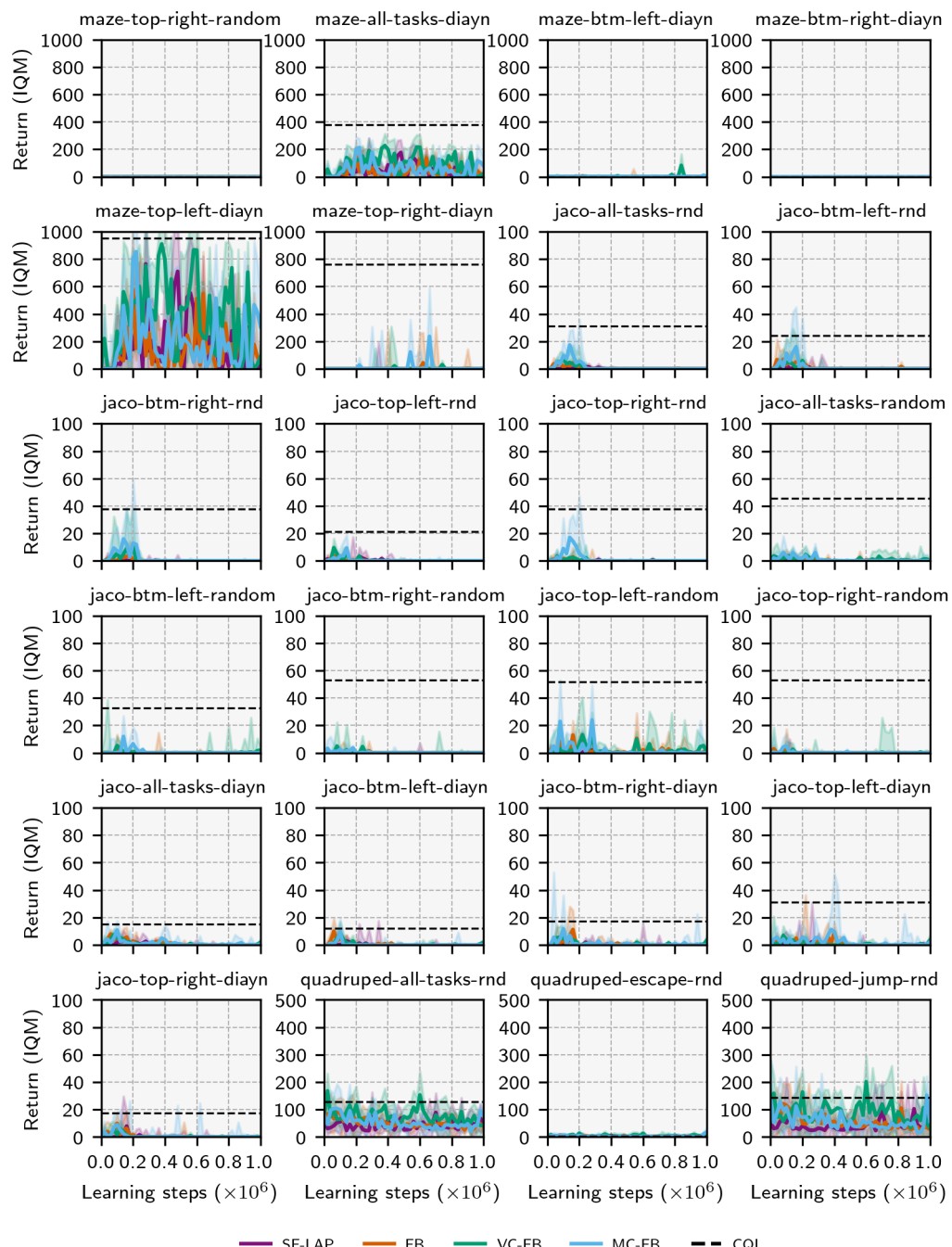

Figure 13: **Learning Curves (2/3)**. Models are evaluated every 20,000 timesteps where we perform 10 rollouts and record the IQM. Curves are the IQM of this value across 5 seeds; shaded areas are one standard deviation.

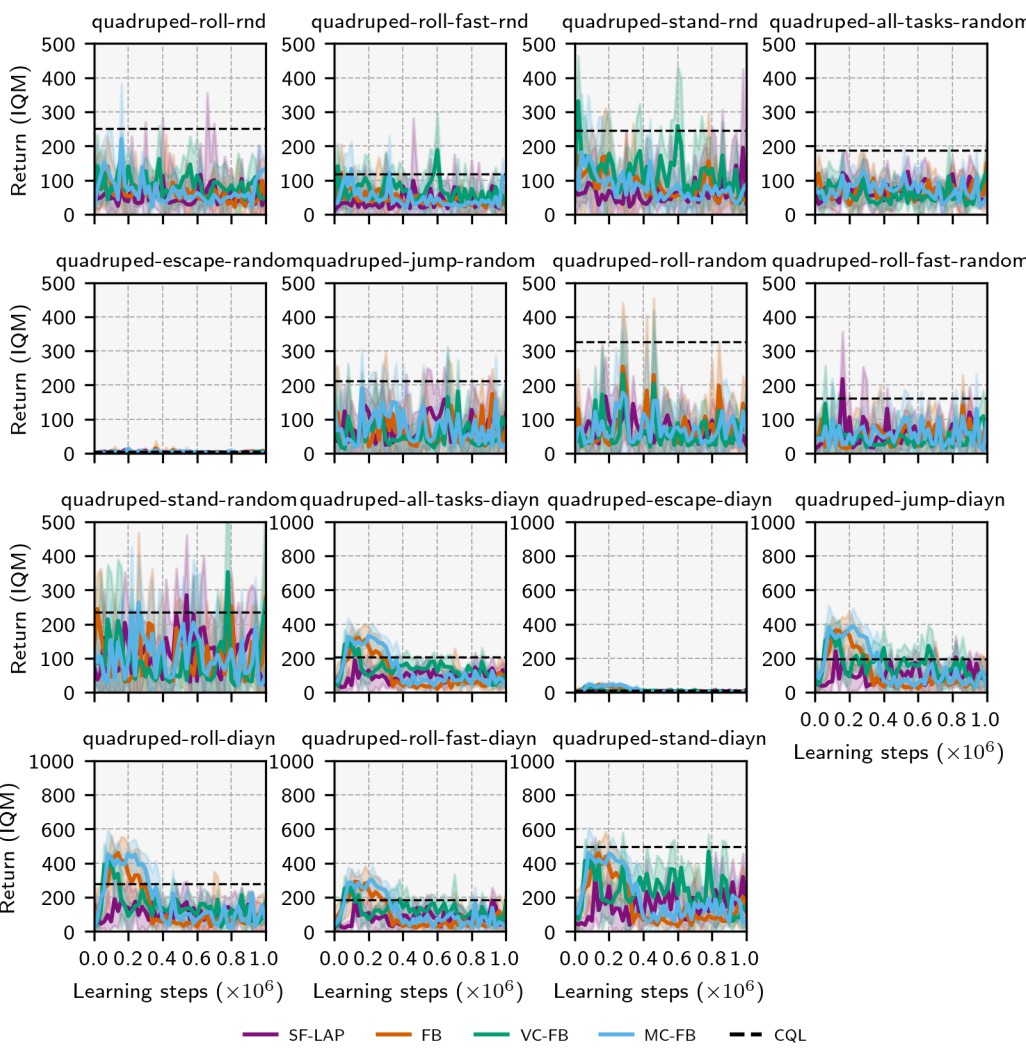

Figure 14: **Learning Curves (3/3)**. Models are evaluated every 20,000 timesteps where we perform 10 rollouts and record the IQM. Curves are the IQM of this value across 5 seeds; shaded areas are one standard deviation.

# I  CODE SNIPPETS

## I.1  UPDATE STEP

```python
def update_fb(
    self,
    observations: torch.Tensor,
    actions: torch.Tensor,
    next_observations: torch.Tensor,
    discounts: torch.Tensor,
    zs: torch.Tensor,
    step: int,
) -> Dict[str, float]:
    """
    Calculates the loss for the forward-backward representation network.
    Loss contains two components:
        1. Forward-backward representation (core) loss: a Bellman update
            on the successor measure (equation 24, Appendix B)
        2. Conservative loss: penalises out-of-distribution actions
    Args:
        observations: observation tensor of shape [batch_size, observation_length]
        actions: action tensor of shape [batch_size, action_length]
        next_observations: next observation tensor of
                           shape [batch_size, observation_length]
        discounts: discount tensor of shape [batch_size, 1]
        zs: policy tensor of shape [batch_size, z_dimension]
        step: current training step
    Returns:
        metrics: dictionary of metrics for logging
    """

    # update step common to all FB models
    (
        core_loss,
        core_metrics,
        F1,
        F2,
        B_next,
        M1_next,
        M2_next,
        _,
        _,
        actor_std_dev,
    ) = self._update_fb_inner(
        observations=observations,
        actions=actions,
        next_observations=next_observations,
        discounts=discounts,
        zs=zs,
        step=step,
    )

    # calculate MC or VC penalty
    if self.mcfb:
        (
            conservative_penalty,
            conservative_metrics,
        ) = self._measure_conservative_penalty(
            observations=observations,
            next_observations=next_observations,
            zs=zs,
            actor_std_dev=actor_std_dev,
            F1=F1,
            F2=F2,
            B_next=B_next,
            M1_next=M1_next,
            M2_next=M2_next,
        )
    # VCFB
    else:
        (
            conservative_penalty,
            conservative_metrics,
        ) = self._value_conservative_penalty(
            observations=observations,
            next_observations=next_observations,
            zs=zs,
            actor_std_dev=actor_std_dev,
            F1=F1,
            F2=F2,
        )

    # tune alpha from conservative penalty
    alpha, alpha_metrics = self._tune_alpha(
        conservative_penalty=conservative_penalty
    )
    conservative_loss = alpha * conservative_penalty

    total_loss = core_loss + conservative_loss

    # step optimiser
```

```
88          self.FB_optimiser.zero_grad(set_to_none=True)
89          total_loss.backward()
90          for param in self.FB.parameters():
91              if param.grad is not None:
92                  param.grad.data.clamp_(-1, 1)
93          self.FB_optimiser.step()
94
95          return metrics
```

## I.2 VALUE-CONSERVATIVE PENALTY

```
1  def _value_conservative_penalty(
2          self,
3          observations: torch.Tensor,
4          next_observations: torch.Tensor,
5          zs: torch.Tensor,
6          actor_std_dev: torch.Tensor,
7          F1: torch.Tensor,
8          F2: torch.Tensor,
9      ) -> torch.Tensor:
10          """
11          Calculates the value conservative penalty for FB.
12          Args:
13              observations: observation tensor of shape [batch_size, observation_length]
14              next_observations: next observation tensor of shape
15                                              [batch_size, observation_length]
16              zs: task tensor of shape [batch_size, z_dimension]
17              actor_std_dev: standard deviation of the actor
18              F1: forward embedding no. 1
19              F2: forward embedding no. 2
20          Returns:
21              conservative_penalty: the value conservative penalty
22          """
23
24          with torch.no_grad():
25              # repeat observations, next_observations, zs, and Bs
26              # we fold the action sample dimension into the batch dimension
27              # to allow the tensors to be passed through F and B; we then
28              # reshape the output back to maintain the action sample dimension
29              repeated_observations_ood = observations.repeat(
30                  self.ood_action_samples, 1, 1
31              ).reshape(self.ood_action_samples * self.batch_size, -1)
32              repeated_zs_ood = zs.repeat(self.ood_action_samples, 1, 1).reshape(
33                  self.ood_action_samples * self.batch_size, -1
34              )
35              ood_actions = torch.empty(
36                  size=(self.ood_action_samples * self.batch_size, self.action_length),
37                  device=self._device,
38              ).uniform_(-1, 1)
39
40              repeated_observations_actor = observations.repeat(
41                  self.actor_action_samples, 1, 1
42              ).reshape(self.actor_action_samples * self.batch_size, -1)
43              repeated_next_observations_actor = next_observations.repeat(
44                  self.actor_action_samples, 1, 1
45              ).reshape(self.actor_action_samples * self.batch_size, -1)
46              repeated_zs_actor = zs.repeat(self.actor_action_samples, 1, 1).reshape(
47                  self.actor_action_samples * self.batch_size, -1
48              )
49              actor_current_actions, _ = self.actor(
50                  repeated_observations_actor,
51                  repeated_zs_actor,
52                  std=actor_std_dev,
53                  sample=True,
54              )  # [actor_action_samples * batch_size, action_length]
55
56              actor_next_actions, _ = self.actor(
57                  repeated_next_observations_actor,
58                  z=repeated_zs_actor,
59                  std=actor_std_dev,
60                  sample=True,
61              )  # [actor_action_samples * batch_size, action_length]
62
63          # get Fs
64          ood_F1, ood_F2 = self.FB.forward_representation(
65              repeated_observations_ood, ood_actions, repeated_zs_ood
66          )  # [ood_action_samples * batch_size, latent_dim]
67
68          actor_current_F1, actor_current_F2 = self.FB.forward_representation(
69              repeated_observations_actor, actor_current_actions, repeated_zs_actor
70          )  # [actor_action_samples * batch_size, latent_dim]
71          actor_next_F1, actor_next_F2 = self.FB.forward_representation(
72              repeated_next_observations_actor, actor_next_actions, repeated_zs_actor
73          )  # [actor_action_samples * batch_size, latent_dim]
74          repeated_F1, repeated_F2 = F1.repeat(
75              self.actor_action_samples, 1, 1
76          ).reshape(self.actor_action_samples * self.batch_size, -1), F2.repeat(
77              self.actor_action_samples, 1, 1
78          ).reshape(
79              self.actor_action_samples * self.batch_size, -1
```

```
80              )
81              cat_F1 = torch.cat(
82                  [
83                      ood_F1,
84                      actor_current_F1,
85                      actor_next_F1,
86                      repeated_F1,
87                  ],
88                  dim=0,
89              )
90              cat_F2 = torch.cat(
91                  [
92                      ood_F2,
93                      actor_current_F2,
94                      actor_next_F2,
95                      repeated_F2,
96                  ],
97                  dim=0,
98              )
99
100             repeated_zs = zs.repeat(self.total_action_samples, 1, 1).reshape(
101                 self.total_action_samples * self.batch_size, -1
102             )
103
104             # convert to Qs
105             cql_cat_Q1 = torch.einsum("sd, sd -> s", cat_F1, repeated_zs).reshape(
106                 self.total_action_samples, self.batch_size, -1
107             )
108             cql_cat_Q2 = torch.einsum("sd, sd -> s", cat_F2, repeated_zs).reshape(
109                 self.total_action_samples, self.batch_size, -1
110             )
111
112             cql_logsumexp = (
113                 torch.logsumexp(cql_cat_Q1, dim=0).mean()
114                 + torch.logsumexp(cql_cat_Q2, dim=0).mean()
115             )
116
117             # get existing Qs
118             Q1, Q2 = [torch.einsum("sd, sd -> s", F, zs) for F in [F1, F2]]
119
120             conservative_penalty = cql_logsumexp - (Q1 + Q2).mean()
121
122             return conservative_penalty
```

## I.3   MEASURE-CONSERVATIVE PENALTY

```
1  def _measure_conservative_penalty(
2          self,
3          observations: torch.Tensor,
4          next_observations: torch.Tensor,
5          zs: torch.Tensor,
6          actor_std_dev: torch.Tensor,
7          F1: torch.Tensor,
8          F2: torch.Tensor,
9          B_next: torch.Tensor,
10         M1_next: torch.Tensor,
11         M2_next: torch.Tensor,
12     ) -> torch.Tensor:
13         """
14         Calculates the measure conservative penalty.
15         Args:
16             observations: observation tensor of shape [batch_size, observation_length]
17             next_observations: next observation tensor of shape
18                                            [batch_size, observation_length]
19             zs: task tensor of shape [batch_size, z_dimension]
20             actor_std_dev: standard deviation of the actor
21             F1: forward embedding no. 1
22             F2: forward embedding no. 2
23             B_next: backward embedding
24             M1_next: successor measure no. 1
25             M2_next: successor measure no. 2
26         Returns:
27             conservative_penalty: the measure conservative penalty
28         """
29
30         with torch.no_grad():
31             # repeat observations, next_observations, zs, and Bs
32             # we fold the action sample dimension into the batch dimension
33             # to allow the tensors to be passed through F and B; we then
34             # reshape the output back to maintain the action sample dimension
35             repeated_observations_ood = observations.repeat(
36                 self.ood_action_samples, 1, 1
37             ).reshape(self.ood_action_samples * self.batch_size, -1)
38             repeated_zs_ood = zs.repeat(self.ood_action_samples, 1, 1).reshape(
39                 self.ood_action_samples * self.batch_size, -1
40             )
41             ood_actions = torch.empty(
42                 size=(self.ood_action_samples * self.batch_size, self.action_length),
43                 device=self._device,
44             ).uniform_(-1, 1)
```

```
45
46              repeated_observations_actor = observations.repeat(
47                  self.actor_action_samples, 1, 1
48              ).reshape(self.actor_action_samples * self.batch_size, -1)
49              repeated_next_observations_actor = next_observations.repeat(
50                  self.actor_action_samples, 1, 1
51              ).reshape(self.actor_action_samples * self.batch_size, -1)
52              repeated_zs_actor = zs.repeat(self.actor_action_samples, 1, 1).reshape(
53                  self.actor_action_samples * self.batch_size, -1
54              )
55              actor_current_actions, _ = self.actor(
56                  repeated_observations_actor,
57                  repeated_zs_actor,
58                  std=actor_std_dev,
59                  sample=True,
60              )  # [actor_action_samples * batch_size, action_length]
61
62              actor_next_actions, _ = self.actor(
63                  repeated_next_observations_actor,
64                  z=repeated_zs_actor,
65                  std=actor_std_dev,
66                  sample=True,
67              )  # [actor_action_samples * batch_size, action_length]
68
69          # get Fs
70          ood_F1, ood_F2 = self.FB.forward_representation(
71              repeated_observations_ood, ood_actions, repeated_zs_ood
72          )  # [ood_action_samples * batch_size, latent_dim]
73
74          actor_current_F1, actor_current_F2 = self.FB.forward_representation(
75              repeated_observations_actor, actor_current_actions, repeated_zs_actor
76          )  # [actor_action_samples * batch_size, latent_dim]
77          actor_next_F1, actor_next_F2 = self.FB.forward_representation(
78              repeated_next_observations_actor, actor_next_actions, repeated_zs_actor
79          )  # [actor_action_samples * batch_size, latent_dim]
80          repeated_F1, repeated_F2 = F1.repeat(
81              self.actor_action_samples, 1, 1
82          ).reshape(self.actor_action_samples * self.batch_size, -1), F2.repeat(
83              self.actor_action_samples, 1, 1
84          ).reshape(
85              self.actor_action_samples * self.batch_size, -1
86          )
87          cat_F1 = torch.cat(
88              [
89                  ood_F1,
90                  actor_current_F1,
91                  actor_next_F1,
92                  repeated_F1,
93              ],
94              dim=0,
95          )
96          cat_F2 = torch.cat(
97              [
98                  ood_F2,
99                  actor_current_F2,
100                 actor_next_F2,
101                 repeated_F2,
102             ],
103             dim=0,
104         )
105
106         cml_cat_M1 = torch.einsum("sd, td -> st", cat_F1, B_next).reshape(
107             self.total_action_samples, self.batch_size, -1
108         )
109         cml_cat_M2 = torch.einsum("sd, td -> st", cat_F2, B_next).reshape(
110             self.total_action_samples, self.batch_size, -1
111         )
112
113         cml_logsumexp = (
114             torch.logsumexp(cml_cat_M1, dim=0).mean()
115             + torch.logsumexp(cml_cat_M2, dim=0).mean()
116         )
117
118         conservative_penalty = cml_logsumexp - (M1_next + M2_next).mean()
119
120         return conservative_penalty
```

## I.4 $\alpha$ TUNING

```python
def _tune_alpha(
    self,
    conservative_penalty: torch.Tensor,
) -> torch.Tensor:
    """
    Tunes the conservative penalty weight (alpha) w.r.t. target penalty.
    Discussed in Appendix B.1.4
    Args:
        conservative_penalty: the current conservative penalty
    Returns:
        alpha: the updated alpha
    """

    # alpha auto-tuning
    alpha = torch.clamp(self.critic_log_alpha.exp(), min=0.0, max=1e6)
    alpha_loss = (
        -0.5 * alpha * (conservative_penalty - self.target_conservative_penalty)
    )

    self.critic_alpha_optimiser.zero_grad()
    alpha_loss.backward(retain_graph=True)
    self.critic_alpha_optimiser.step()
    alpha = torch.clamp(self.critic_log_alpha.exp(), min=0.0, max=1e6).detach()

    return alpha
```

