# OpenReview forum: "Conservative World Models"
_ICLR.cc/2024/Conference — Submitted to ICLR 2024_

### Official Review · Reviewer_5Tki · 2023-11-01

**Soundness:** 2 fair
**Presentation:** 3 good
**Contribution:** 2 fair
**Rating:** 3
**Confidence:** 4

**Summary:**

This paper introduces the integration of conservative terms into the Forward-Backward (FB) loss function, aiming to mitigate the risk of overvaluing out-of-distribution state-action pairs—a factor that could significantly impair FB’s performance. The empirical results presented in the study effectively validate the utility of these conservative terms.

**Strengths:**

- The rationale behind implementing conservative terms is well-founded and convincing.
- The experimental evidence provided clearly demonstrates the efficacy of the conservative terms.
- A didactic example is skillfully used to elucidate the impact and functionality of the conservative terms.

**Weaknesses:**

- The “world model” as described in the paper seems to be potentially mischaracterized. The referenced document, [2209.14935.pdf (arxiv.org)](https://arxiv.org/pdf/2209.14935.pdf), elucidates that "Both SFs and FB lie in between model-free and model-based RL, by predicting features of future states, or summarizing long-term state-state relationships. Like model-based approaches, they decouple the dynamics of the environment from the reward function. Contrary to world models, they require neither planning at test time nor a generative model of states or trajectories." Therefore, the paper’s description might need a revision for accuracy.
- The set of baselines employed in the experimental section appears to be inadequate. As depicted in Figure 2 of [2209.14935.pdf (arxiv.org)](https://arxiv.org/pdf/2209.14935.pdf), FB does not have the "sufficient performance gap" as claimed by the authors, necessitating the inclusion of a broader spectrum of baselines for a more comprehensive performance comparison.
- The concept presented is relatively straightforward, essentially adapting the conservative term from Conservative Q-Learning (CQL) in offline reinforcement learning to the zero-shot reinforcement learning context. Given the moderate domain gap between offline RL and zero-shot RL, the idea appears to be somewhat lacking in novelty.

**Questions:**

- Could the authors revisit and verify the usage of the term "world model" in the manuscript? The source [2209.14935.pdf (arxiv.org)](https://arxiv.org/pdf/2209.14935.pdf) suggests a possible misclassification of FB, and by extension, the methodology in this paper, as a "world model".
- The term "world model" seems to have limited relevance and is infrequently used throughout the paper. Could its significance to the paper’s core content be clarified, or is it a concept that could potentially be omitted without loss of clarity?
- To bolster the robustness of the study, it would be beneficial for the authors to incorporate a wider array of zero-shot RL baselines, particularly those utilized in [2209.14935.pdf (arxiv.org)](https://arxiv.org/pdf/2209.14935.pdf). Additionally, considering the 2022 inception of the FB method, it may be pertinent to include more recent and relevant baselines in the analysis.

---

> ### Author Response · Authors · 2023-11-14
> **Author Response to Reviewer 5Tki**
>
> We thank the reviewer for their engagement. We note significant disagreements in our shared perceptions of the paper, and are keen to engage with the reviewer to find common ground. We respond to their queries below.
>
> **Q1: Could the authors revisit and verify the usage of the term "world model" in the manuscript? The source [2209.14935.pdf (arxiv.org)](https://arxiv.org/pdf/2209.14935.pdf) suggests a possible misclassification of FB, and by extension, the methodology in this paper, as a "world model"**
>
> See global response W1.
>
> **Q2:** **The set of baselines employed in the experimental section appears to be inadequate. As depicted in Figure 2 of [2209.14935.pdf (arxiv.org)](https://arxiv.org/pdf/2209.14935.pdf), FB does not have the "sufficient performance gap" as claimed by the authors, necessitating the inclusion of a broader spectrum of baselines for a more comprehensive performance comparison.**
>
> We disagree that FB does not sufficiently outperform other SF-based, zero-shot RL methods. Touati et. al (2022) show FB achieved an aggregate score ~10% higher than the SOTA SF-based method after extensive experimental analysis (Figure 1 (an aggregation of Figure 2) and Table 2 in [1]).  FB arguably enjoys better theoretical properties as it avoids the performance ceiling induced by the representation learning methods required for SF-based methods. That said, our proposals are fully compatible with SF-based methods, and they too will suffer from the same failure mode as FB on low quality datasets (TD learning with $a_{t+1} \sim \pi(s_{t+1})$--see Equation 6 in [1]). As such we shall add a new appendix that derives losses for training Conservative Successor Features—see Revision 5. Focussing new zero-shot RL work on FB but showing compatibility with SF in this way has been shown in other works (under review at ICLR) [2].
>
> **Q3: Additionally, considering the 2022 inception of the FB method, it may be pertinent to include more recent and relevant baselines in the analysis.**
>
> To the best of our knowledge, we are the first work to build upon FB representations in the context of zero-shot RL. If the reviewer is aware of works we have missed we ask for them to be shared so we can include them.
>
> ### References
>
> [1] Touati, Ahmed, Jérémy Rapin, and Yann Ollivier. "Does Zero-Shot Reinforcement Learning Exist?." *arXiv preprint arXiv:2209.14935* (2022).
>
> [2] https://openreview.net/forum?id=qnWtw3l0jb

---

> > ### Author Response · Authors · 2023-11-20
> > **Comment to Reviewer 5Tki after Revision One**
> >
> > We wanted to let you know that our updated manuscript should address all of the concerns highlighted in your review.
> >
> > **Q1.** We have retitled the paper and dropped any reference to world models.
> >
> > **Q2.** We have provided an additional baseline: the state-of-the-art SF based method from Touati et al (laplacian eigenfunctions). As expected it performs worse than FB (with the performance gap consistent with that reported by Touati et al) and as a consequence our conservative variants.
> >
> > **Q3.** There are no other FB-based baselines to include.
> >
> > We hope these revisions alleviate your main concerns. Please do let us know if you have any further questions!

---

> > > ### Author Response · Authors · 2023-11-22
> > > **Comment to Reviewer 5Tki after Final Revision**
> > >
> > > We have uploaded our final revision of the manuscript. We are disappointed that you have chosen not to participate in the discussion period. Nonetheless, we have made every effort to engage, and have either addressed all of your concerns via comment or via new experiments. We would really appreciate it if you updated your score in light of our comments and updates. Thanks very much in advance for taking the time to do so.

---

### Official Review · Reviewer_xRo3 · 2023-11-04

**Soundness:** 3 good
**Presentation:** 3 good
**Contribution:** 2 fair
**Rating:** 5
**Confidence:** 3

**Summary:**

In this paper the authors propose what they call a "conservative world model", essentially a general model that can demonstrate zero-shot RL through offline data. The conservative mentioned in the title refers to an approach to be more conservative with the values of out-of-distribution state-action pairs, which can allow for more general policies in some cases. The authors cover their approach in detail, present two variations of their approach and compare their performance to a non-conservative baseline across four standard domains. Their results demonstrate comparable or better performance with the non-conservative baseline.

**Strengths:**

This paper demonstrates a fair amount of clarity around the approach, with well-written arguments around the potential benefits. The work is original, to the best of my knowledge, in terms of its weighting approach for the training data for offline RL problems. The quality of the work in terms of the number of domains and overview of the results if good. Those interested in offline RL, and particular RB approaches, will likely find the work of some significance.

**Weaknesses:**

This paper has a number of weaknesses.

First, a relatively minor one is the choice of the phrase "world model" to describe the approach. World models already are a well-established and distinct approach in this area [1].

Second is the relative lack of novelty. The approach is essentially a reweighing of training data, but similar reweighing strategies have already been proposed and are not compared against [2].

Third is the evaluation setup. It's odd to have two variants of the approach in comparison to a single baseline, which essentially gives the approach twice the opportunity to outperform the baseline. For simplicity, it might have been better to stick with VC-FB and leave MC-FB to the appendix. However, it's also unfair that both approaches are given 3 times the training duration to FB. Ablations or additional baselines could have helped to avoid this issue.

Fourth is the presentation of the results. The paper repeatedly aggregates over distinct problems, whose scores are not comparable and presents these results as summed values and percentages. I don't believe this is appropriate. The claims made are also not reflective of the results, which show an inconsistent improvement by VC-FB and MC-FB over FB. This is especially worrying in the Random case, where the approaches are essentially identical. Given the claims around the value of conservative world models, I would have assumed that they would have outperformed FB the most when the dataset was of a poorer quality.

[1] Ha, David, and Jürgen Schmidhuber. "World models." arXiv preprint arXiv:1803.10122 (2018).
[2] Robine, Jan, et al. "Transformer-based World Models Are Happy With 100k Interactions." arXiv preprint arXiv:2303.07109 (2023).

**Questions:**

1. Is there a relationship to the more typical usage of World Model that I'm missing?
2. Are there no other appropriate baselines that could have been included in the evaluation?
3. Why is it appropriate to aggregate the results across evaluation domains?
4. Why would the authors' approach perform worst with worse datasets?

---

> ### Author Response · Authors · 2023-11-14
> **Author Response to Reviewer xRo3**
>
> We thank the reviewer for their engagement. We note significant disagreements in our shared perceptions of the paper, and are keen to engage with the reviewer to find common ground. We respond to their queries below.
>
> **Q1: Is there a relationship to the more typical usage of World Model that I'm missing?**
>
> See global response W1.
>
> **Q2: Are there no other appropriate baselines that could have been included in the evaluation?**
>
> Touati et. al show FB achieved an aggregate score ~10% higher than the SOTA SF-based method after extensive experimental analysis (Figure 1 and Table 2 in [1]). Indeed, FB arguably enjoys better theoretical properties as it avoids the performance ceiling induced by the representation learning methods required for SF-based methods. For these reasons we believe FB is the SOTA zero-shot RL method and use it as our sole baseline. Are there specific zero-shot RL baselines the reviewer would like us to include?
>
> **Q3: Second is the relative lack of novelty. The approach is essentially a reweighing of training data, but similar reweighing strategies have already been proposed and are not compared against [2].**
>
> We disagree that our approach is reweighting of the training data and therefore lacks novelty. We believe the reviewer has misunderstood our proposals; we are not reweighting the training data, we are (in the case of VC-FB) regularising Q functions to mitigate OOD value overestimation—data is sampled uniformly from dataset $\mathcal{D}$. The method proposed in the cited paper is not a zero-shot RL method (the policy cannot generalise to arbitrary tasks in an environment with zero downstream planning or learning) and so we do not believe it would constitute a valid baseline.
>
> **Q4: It's odd to have two variants of the approach in comparison to a single baseline, which essentially gives the approach twice the opportunity to outperform the baseline. For simplicity, it might have been better to stick with VC-FB and leave MC-FB to the appendix. However, it's also unfair that both approaches are given 3 times the training duration to FB. Ablations or additional baselines could have helped to avoid this issue.**
>
> Firstly, we note that we compare against 3 baselines not 1—see Table 5 Appendix C. Secondly, we proposed two variants of our method because it was not clear _a priori_ which method of $z$ sampling would prove beneficial. We do not think this constitutes twice the opportunity to outperform our baselines. As it transpired, both methods outperformed all of our baselines in aggregate on their own merit. Thirdly, and finally, we hold the number of update steps and training data constant for all algorithms which is standard practice. The reason training our proposals takes 3x longer than existing methods is not because the methods were exposed to more data or learning steps, but rather because the $Q$-function regularisation invokes expensive operations. We discuss this in Section 5, and suggest this could be resolved with thoughtful engineering of the codebase. Doing so was left for future work.
>
> **Q4: Why is it appropriate to aggregate the results across evaluation domains?**
>
> DeepMind Control domains are specifically designed such that task scores always lie in $[0, 1000]$ and so are directly comparable. Aggregating across these domains is standard practice in the community, we refer the reviewer to e.g. [1, 2] for examples. We aggregate in the same way as the work we build upon [3].
>
> **Q5: Why would the authors' approach perform worst with worse datasets?**
>
> See global response W2.
>
> ### References
>
> [1] Fujimoto, Scott, and Shixiang Shane Gu. "A minimalist approach to offline reinforcement learning." *Advances in neural information processing systems* 34 (2021): 20132-20145.
>
> [2] Kostrikov, Ilya, Ashvin Nair, and Sergey Levine. "Offline reinforcement learning with implicit q-learning." *arXiv preprint arXiv:2110.06169* (2021).
>
> [3] Touati, Ahmed, Jérémy Rapin, and Yann Ollivier. "Does Zero-Shot Reinforcement Learning Exist?." *arXiv preprint arXiv:2209.14935* (2022).
>
> [4] https://openreview.net/forum?id=qnWtw3l0jb

---

> > ### Comment · Reviewer_xRo3 · 2023-11-14
> > **Re: Author Response to Reviewer xRo3**
> >
> > For Q1, I would encourage the authors to change the term as the community does appear to be consolidating around world models as meaning a distinct modelling problem.
> >
> > For Q2, looking at Touati et. al's work, the offline variant of their approach was outperformed for quadruped_stand with comparable performance in several other environments. As such, it seems fair to speculate that these baselines would not necessarily be strictly dominated.
> >
> > For Q3, I apologize for mischaracterizing the work and appreciate the clarification! Given Reviewer 5Tki's concerns I am still concerned about the novelty, however.
> >
> > For Q4a, the inclusion of two variants of the approach, without any a priori knowledge, would still give 2-1 odds of outperforming a baseline. I apologize for missing the other baselines from the supplementary materials!
> >
> > For Q4b, I apologize, I was simply unfamiliar with this being a standard practice.
> >
> > For Q5, I appreciate the speculation but I am still concerned by these results. Without additional results it becomes difficult to determine the cause here.
> >
> > As I still have concerns around the novelty and value (via results) of the work, I am not changing my score at this time.

---

> > > ### Author Response · Authors · 2023-11-20
> > > **Comment to Reviewer xRo3 after Revision One**
> > >
> > > Thanks for your quick response above, we really appreciate it. Below we explain how we’ve addressed your concerns in the updated manuscript.
> > >
> > > **Q1.** We’ve changed the title in our revised manuscript and removed all mention of world models.
> > >
> > > **Q2.** We implemented the state-of-the-art SF based method from Touati et al (Laplacian eigenfunctions). As expected it performs worse than FB (with the performance gap consistent with that reported by Touati et al) and as a consequence our conservative variants. As noted in the global response, and outlined via our derivations in Appendix E, our methods still make sense with SF based methods if one wanted to use them in that context. We thank the reviewer for pushing us to implement the new baseline.
> > >
> > > **Q5.** On the full datasets the largest performance gap between FB and our conservative variants was on the lowest quality dataset (Random). We believe this corroborates our hypothesis that the Random-100k dataset is too uninformative to learn good models/polices from, and addressed your question around why relative performance was worst on the lowest quality dataset.
> > >
> > > We hope to have now addressed your primary concerns, please do let us know if you have any further queries!

---

> > > > ### Comment · Reviewer_xRo3 · 2023-11-21
> > > > **Re: Comment to Reviewer xRo3 after Revision One**
> > > >
> > > > Thanks for the further updates! My concerns in terms of baselines (Q2) and the interpretation of the results (Q5) have been addressed. I am still concerned with the novelty issue brought up by 5Tki "The concept presented is relatively straightforward, essentially adapting the conservative term from Conservative Q-Learning (CQL) in offline reinforcement learning to the zero-shot reinforcement learning context. Given the moderate domain gap between offline RL and zero-shot RL, the idea appears to be somewhat lacking in novelty."
> > > >
> > > > I have raised my rating, but would appreciate the author's response to the above quote from 5Tki.

---

> > > > > ### Author Response · Authors · 2023-11-21
> > > > > **Re: xRo3**
> > > > >
> > > > > Thanks again for your quick response and for updating your score. We would take issue with 5Tki’s claim that there is a “moderate domain gap” between offline RL and zero-shot RL.  Traditional offline RL methods simply learn one value function / policy to solve one task; zero-shot RL methods learn an infinite set of value functions / policies to solve any arbitrary task. Zero-shot RL methods are necessarily very different from offline RL methods as a consequence. Although training zero-shot RL methods with constraints may seem intuitive, we are the first to do so. It was not at all obvious to us that such constraints would translate to consistent zero-shot performance improvements.
> > > > >
> > > > > Please let us know if you have any further questions.

---

> > > > > > ### Author Response · Authors · 2023-11-22
> > > > > > **Comment to Reviewer xRo3 after Final Revision**
> > > > > >
> > > > > > We have uploaded our final revision of the manuscript and, as the end of the discussion period nears, we wanted to summarise the changes we’ve made in response to your feedback. These include:
> > > > > >
> > > > > > - A new baseline (SF with Laplacian Eigenfunctions).
> > > > > > - Title change and any reference to world models
> > > > > > - Experiments on full datasets to explain performance discrepencies across datasets
> > > > > >
> > > > > > We believe we have also answered all other queries you’ve raised. Since you have raised no further concerns or asked for any further clarifying information, we hope you feel happy to update your score as a consequence. Thanks very much again for your feedback.

---

### Official Review · Reviewer_85iy · 2023-11-07

**Soundness:** 2 fair
**Presentation:** 3 good
**Contribution:** 3 good
**Rating:** 5
**Confidence:** 3

**Summary:**

This paper proposes two novel reinforcement learning methodologies for learning arbitrary tasks from unsupervised transition data. The key is that the datasets are not as large and diverse as in previous work. The goal is to pre-train a model on small transition datasets and at test time, zero-shot generalize to arbitrary reward functions that are characterized by some distribution. The work builds on the idea of forward-backward learning in which successor information is employed to make statements about past and future visitation counts of a policy. These counts are then related to reward information during test time. The proposed algorithms use conservativeness to combat large Q-value predictions in unseen state-action areas which is an idea that has previously been used successfully in standard offline RL. The work validates the functionality of the method with an intuitive toy example and then goes on to evaluate on larger benchmark suites where the proposed methods outperform baselines on several tasks.

I would like to mention that while I am aware of the offline RL literature, I am not familiar with the forward-backward literature specifically. I did not check the math in section $2$ for correctness since it is already published work.

**Strengths:**

Motivation
* The motivation of the work is clearly established. The ability to extract dynamics information that can be leveraged for arbitrary downstream tasks is a promising direction towards building general-purpose representations. The reasoning for the employment of forward-backward methods is clearly established.

Contextualization with prior work
* Specifically, the first few sections do a good job of making sure that prior work is mentioned and highlight clearly which parts of the manuscript are novel and which ideas are taken from previous work.

Structural clarity
* The paper has a very clear structure and is easy to follow. The origins of the method are clear and the toy example helps understand the internal mechanisms of the network. The language is clear and the manuscript is well-written.

Limitations section
* The work provides a good limitations section that highlights some of the practical challenges and outlines future directions of work. In general, I think this section is very beneficial for readers of the paper and I think it is a nice addition.

Experimental Evaluation
* The experimental section is structured well and asks $3$ very relevant questions to analyze the proposed algorithms. Specifically, experiment Q3 in the paper shows that in most cases it is likely not bad to simply add conservativeness to the existing approach as long as the data quality is high is convincing.
* The benchmark suite is sufficiently sized and provides several different types of environments.

Novelty
* As mentioned before, I’m not familiar with this specific type of model but given the recency of publication of prior methods, it seems reasonable to assume that the contribution is sufficiently novel. I also think the contribution adds sufficient new content to the existing approach and highlights the relationship between conservative offline RL approaches and how the procedures transfer to forward-backward methods.

**Weaknesses:**

Mathematical clarity
* Specifically section 2 might benefit from clarity improvements and re-ordering of some of the references.
  * The relationship between the task structure of the set of MDPs and the reward functions is unclear. The MDP framework outlined in section $2$ paragraph $1$ does not come with such structure and the structure is only mentioned but not well-defined in the problem formulation section. As a result, it is not clear to me where the differences in the sets of tasks come from (see Q1). It might be beneficial to mention the distribution of task vectors earlier and how it relates to reward functions in paragraph $2$ (Problem formulation.).
  * Relatedly, should the forward model function be defined as $F: \mathcal{S} \times \mathcal{A} \times \mathcal{Z} \mapsto \mathbb{R}^d$? Do the tasks and the policies share the same space? Is the embedding space always the size of the task vector space? In this context, it is not clear what the following notation means: $\pi_z : \mathcal{z} ∼ \mathcal{z}$.
  * I was looking for a reference for the derivation of some of the claims in section $2$ and that reference is only provided at the end of the section. I personally would have benefitted from this being at the beginning. This would make it both clear that the derivation comes from previous work and possible for the reader to open up the reference on the side before I dive into the math.
  * The actor-critic formulation is only mentioned very briefly at the end of the section but as far as I understand it might be quite crucial to running the experiments and deriving the actual method.

Contextualization with prior work
* There is an abundance of literature on conservativeness in RL and the paragraph on this in the related works section contains a total of 7 references. This contextualization could be a little stronger. The downside of this lack of contextualization shows somewhat in the experiments as I will outline later. Less detail about the cited methods and concise statements about previous literature’s commonalities might provide a way to condense the text in this section.

Experimental Evaluation
* The evaluation metrics are not fully specified. The text mostly talks about performance but it is not clear what is being measured. I’m going to assume that the experiments measure performance in terms of cumulative reward. Relatedly, in Figure 4, It is not clear to me what a Task Score x is.
* The following statement is probably not well-qualified. “CQL is representative of what a conservative algorithm can achieve when optimizing for one task in a domain rather than all tasks”. CQL was an impressive step towards using offline data in RL but since then many more conservative methods have been established that show significantly stronger results but even more importantly are easier to train and more stable. This is one of the weaknesses of CQL (and the proposed method) that is highlighted in section 5 and I strongly agree. As a result, CQL might not be the best baseline to establish that the method performs as well as common offline RL methods. It might make sense to compare to an offline method that provides better stability. A common choice would probably be Implicit Q-Learning. That being said, CQL is a reasonable choice in the sense that it tries to achieve a similar objective as the proposed method and as such provides a comparison against a non-forward-backward approach with similar mechanistics. Still, there might be better baselines to strengthen the claim that the method competes with sota offline RL approaches.
* Some of the claims may be overstated and the text could be a little more detailed on the actual findings.
  * While I think the idea of reporting percentage improvements can be nice to establish a clear performance difference, this difference is only really strong if the baseline performance is already good. I could, for instance, say that the method performs $1000%$ better on the Jaco RND experiment than the non-conservative baseline. However, this would be rather misleading because neither method might be close to solving the task. Depending on how these measures are now aggregated, the performance of the method might look inflated. I think a clear definition of what’s being measured and how the measures are aggregated would be useful to provide context for the numbers and a more detailed description of when the method works well.
  * The claim that conservativeness is not harmful is only supported on what is referred to as a high-quality dataset. In this case, the dataset is already providing decent coverage and the Q-values should be easy to approximate (see Q3).
  * The difference between VC-FB and MC-FB is highlighted in section $3$ but the effects of the differences are not explicitly analyzed in the experiments.

Minor textual clarity suggestions
* Figure 1’s caption could probably mention earlier that this is not simply an illustration but the data is from an actual experiment.
* Equation $(1)$ is currently not an expectation without assumptions on the distribution.
* The statement “Since the successor measure satisfies a Bellman equation” would benefit from a citation.
* “$s_+$ is sampled independently from $(s_t, a_t, s_{t+1})$”, the latter is not a distribution but a tuple, this should probably be the dataset?

Overall, I believe that the paper establishes that there are benefits to applying conservative concepts to the forward-backward style method that is employed. The methods seem to be easy enough to implement, in general not to do worse on the proposed baselines and improve upon the baseline in several instances. However, I do think section $2$ would benefit from a few clarity improvements and that the reporting and textual analysis of the experiments could be a little more detailed. I’d be happy to raise my score if these concerns are addressed.  Additionally, experiments with a stronger offline RL baseline might make a stronger case but are not necessarily required for me to raise my score.

**Questions:**

Q1: How is the task distribution $\mathcal{Z}$ defined in practice? It is not quite clear to me, for instance, how I would define such an abstract distribution for either of the given environments in section $5$ other than by possibly defining a distribution over final states which would have to be as large as the state space? Can you give an example of what this would look like in, e.g. the Cheetah environment?

Q2: Traditional CQL samples the data it’s minimizing its Q-values over from the policy distribution rather than employing a max operator. What is the reasoning for choosing a max operator here rather than just sampling from the distribution? The latter seems to be easier to implement.

Q3: What do we expect to happen when we run the conservative version of the proposed algorithm on full datasets with poor quality? Are the trends similar to what we see in the small-scale experiments?

Q4: Is it possible to deduce up-front when this approach performs as well as or better than its single-task offline RL counterparts? In other words, what types of environment properties make the method work?

Q5: It is a little counterintuitive to me that conservativeness seems to work better when the datasets are of higher quality rather than when datasets provide poor support. Do you have any idea why that is?

Q6: (Feel free not to answer since this is a question about previous work rather than your work really.) Are $F$ and $B$ learned solely through equation (4) in the original FB approach? How can we be sure they are actual distribution summaries then?

---

> ### Author Response · Authors · 2023-11-14
> **Author Response to Reviewer 85iy (1/2)**
>
> We thank the reviewer for engaging with our manuscript and providing useful feedback. We summarise your comments/questions and respond to them below.
>
> **Q1: How is the task distribution $Z$ defined in practice? It is not quite clear to me, for instance, how I would define such an abstract distribution for either of the given environments in section 5 other than by possibly defining a distribution over final states which would have to be as large as the state space? Can you give an example of what this would look like in, e.g. the Cheetah environment?**
>
> We discuss the formulation of $\mathcal{Z}$ in Appendix B.1.2, but note, and apologise, that a reference to this appendix in the main body was missed. We address will this in Revision 4.3. We believe further misunderstanding was likely caused by our presentation of the FB theory which we address in Revision 2.
>
> **Q2: Traditional CQL samples the data it’s minimizing its Q-values over from the policy distribution rather than employing a max operator. What is the reasoning for choosing a max operator here rather than just sampling from the distribution? The latter seems to be easier to implement.**
>
> Traditional CQL does indeed employ a max operator over the current Q iterate in theory (see red text in Equation 3 in [1]), which is then approximated with samples from both the policy distribution conditioned on the current observation, and a uniform distribution (Appendix F in [1]). The official CQL implementation also samples from the policy distribution conditioned on the next observation [2], though this is not discussed in their paper. Our method for approximating the max operation outlined in Appendix B.1.3 is fully consistent with this framework.
>
> **Q3: What do we expect to happen when we run the conservative version of the proposed algorithm on full datasets with poor quality? Are the trends similar to what we see in the small-scale experiments?**
>
> This is an interesting question which we are exploring with further experiments. See Revision 1.
>
> **Q4: Is it possible to deduce up-front when this approach performs as well as or better than its single-task offline RL counterparts? In other words, what types of environment properties make the method work?**
>
> This is an important question that we had hoped Figure 5 could help answer, but we struggled make concrete inferences, and therefore only provide speculative responses here. It appears that zero-shot methods are more likely to outperform the single-task baselines on locomotion tasks (VC-FB > CQL on 4/6 Walker/Quadruped datasets) than on goal-reaching tasks (VC-FB > CQL on 1/6 Maze/Jaco datasets). It seems likely that the absolute performance of the zero-shot methods on evaluation tasks correlates with how likely they were to be sampled from $\mathcal{Z}$ during training. Perhaps the current instantiation of $\mathcal{Z}$ is a better approximation of the locomotion task space. We attempted make t-SNE visualisations of $z$-space to explore this, but found it difficult to draw firm conclusions ($z$-space is 50-dimensional). Further investigation of the make-up of z-space as discussed in Section 5 and should help us get a better grip on this. This is difficult, but important future work. We shall add text to this end—see Revision 4.8.
>
> **Q5: It is a little counterintuitive to me that conservativeness seems to work better when the datasets are of higher quality rather than when datasets provide poor support. Do you have any idea why that is?**
>
> As mentioned in W2 of the global response, we suspect that the combination of low diversity and small absolute size of the Random dataset makes model/policy inference particularly tricky. We hope the proposed experiments on the full Random and DIAYN datasets (Revision 1) will shed light on this hypothesis.
>
> **Q6: Contextualization with prior work: There is an abundance of literature on conservativeness in RL and the paragraph on this in the related works section contains a total of 7 references. This contextualization could be a little stronger. The downside of this lack of contextualization shows somewhat in the experiments as I will outline later. Less detail about the cited methods and concise statements about previous literature’s commonalities might provide a way to condense the text in this section.**
>
> We agree that the related work section on Offline RL could provide more detail, and as a result will add several more references. Namely: IQL [3], PEVI, [4], COMBO [5] and [6]

---

> > ### Author Response · Authors · 2023-11-14
> > **Author Response to Reviewer 85iy (2/2)**
> >
> > **Q7: The evaluation metrics are not fully specified. The text mostly talks about performance but it is not clear what is being measured. […] I think a clear definition of what’s being measured and how the measures are aggregated would be useful to provide context for the numbers and a more detailed description of when the method works well.**
> >
> > Agreed. See Revision 4.4.
> >
> > Finally, we thank the reviewer provided feedback on textual clarity, these will be incorporated in Revision 4.5.
> >
> >
> > ###  References
> >
> > [1] Kumar, Aviral, et al. "Conservative q-learning for offline reinforcement learning." *Advances in Neural Information Processing Systems* 33 (2020): 1179-1191.
> >
> > [2]https://github.com/aviralkumar2907/CQL/blob/d67dbe9cf5d2b96e3b462b6146f249b3d6569796/d4rl/rlkit/torch/sac/cql.py#L241
> > [3] Kostrikov, Ilya, Ashvin Nair, and Sergey Levine. "Offline reinforcement learning with implicit q-learning." *arXiv preprint arXiv:2110.06169* (2021).
> >
> > [4] Jin, Ying, Zhuoran Yang, and Zhaoran Wang. "Is pessimism provably efficient for offline rl?." *International Conference on Machine Learning*. PMLR, 2021.
> >
> > [5] Yu, Tianhe, et al. "Combo: Conservative offline model-based policy optimization." *Advances in neural information processing systems* 34 (2021): 28954-28967.
> >
> > [6] Xie, Tengyang, et al. "Bellman-consistent pessimism for offline reinforcement learning." *Advances in neural information processing systems* 34 (2021): 6683-6694.

---

> > > ### Comment · Reviewer_85iy · 2023-11-18
> > >
> > > I appreciate the clarifications. Thank you.
> > >
> > > **Wrt. Q2**: I should have been more specific. CQL does not employ a max operator over actions but over the regularization distribution. Actions are then sampled from this regularization distribution (see Eq 3 in [1] as you pointed out). When the max operator over actions $a$ is selected, a specific instantiation of the distribution $\mu$ is chosen. I was wondering why that is and how this max over a continuous distribution is computed. This choice intuitively seems unstable but I could be wrong. However, the referenced as well as the submitted code bases seem to be doing something different than the equations in the text since they are actually sampling from the Actor. It might make sense to be more precise here to avoid future confusion.
> > >
> > > **Wrt to title choice**: I agree with other reviewers that a world model in the context of RL can be seen as something different. The early model-based deep RL literature will refer to a world-model as a model that given some state and action does unsupervised training to obtain information about the world. More often than not it will refer to a model that can predict a future state and reward.  However, it seems that recently this term has been used more loosely. I did find that title more appropriate than foundation model. A foundation model nowadays is most likely going to refer to a large-scale generative model trained on vast amounts of unsupervised data.
> > >
> > > It seems that there is in fact a form of unsupervised representation learning over states happening in the code. While not exactly what previous literature would expect, it's certainly closer to a world than foundation model in my opinion. However, this loss is not really mentioned in section 2 which may have led to confusion. I think a revision of section 2 could make this more clear because the key unsupervised term that is used in the code but never explained in the text could be added.
> > >
> > > Personally, I think both titles are ambiguous and don't convey much information but I also don't mind too much. Yet, it is not quite clear to as to why they were chosen and my feedback is that I expected something different when I read the title of this paper. Similarly, I would expect something *very* different if I were to read the words "foundation model" in the title.

---

> > > > ### Author Response · Authors · 2023-11-20
> > > > **Comment to Reviewer 85iy after Revision One**
> > > >
> > > > Thank you very much for further feedback!
> > > >
> > > > **Q2.** We found the notation used in Kumar et. al (2020) unintuitive and so tried to adapt it for legibility. Clearly this didn’t work, and you were right to question our notation. In the updated manuscript we have realigned notation in Equations 7, 8 and 9 to match that used in Kumar et. al (2020) i.e. we’ve removed the $\max_a$ and replaced with the regularisation distribution $\mu(a|s)$. We have also fleshed out Appendix B.1.3 to explain how one gets from Equation 7 to the sampling actions from the actor as you observed in the official CQL codebase and our codebase. We hope this is now more legible and answers your questions.
> > > >
> > > > **Title.** Thanks too for your feedback on the proposed re-title. We provide an updated title in the revision manuscript that we hope strikes a better balance between your feedback and that provided by other reviewers.
> > > >
> > > > Please do let us know if you have any further questions! Thanks again.

---

> ### Comment · Reviewer_85iy · 2023-11-22
>
> Thank you again for the clarifications, I went through the revision of the paper, read other reviewers concerns and comments as well as the corresponding responses.
>
> Wrt. my initial review the following points have been resolved.
> * The concerns for mathematical clarity have mostly been alleviated. I still think $F$ should probably be defined differently but that's just nitpicking at this point. Section 2 is much clearer now.
> * The contextualization with prior work is much stronger than before. I appreciate this.
> * Clarifications on experimental settings and claims are implemented.
> * "conservativeness is not harmful" - claim has been strengthened
>
> The following points remain open
> * The introduction of two methods likely warrants an in-depth comparison between the two to explain their behavior. This seems to be a shared perception across multiple reviewers. I appreciate the conjecture in section 5 and think it is promising. As one of the responses mentions, this seems to be a testable hypothesis and I believe that an experiment verifying this should probably appear in a manuscript like the one presented.
> * I'm still confused about how a task is presented to the agent. I'm guessing a task being revealed to the agent (see section 2) means it is presented with a $z$ variable. It is still not clear to me, what exactly $z$ represents in the current tasks concretely or how this task is presented to the agent at test time. My question about an example was not addressed. I appreciate the clarification on sampling $z$ in the model space (i.e. the training procedure) in Appendix B.1.2. However, my confusion was mostly focused on the environment space during execution. This might help characterize how different objectives are from the data support.
> * It still stands that the method seems to have statistically significant improvements on only 3/12 tasks which is not conveyed adequately. There are actually cases where regularization decreases performance even if in the limit of data on average it does not.
>
> Minor formatting point:
> * Equation (9) goes into the margin
>
> I think this paper has improved on various fronts but still has multiple areas of improvement in the experimental section. Specifically,
> a) characterizing how a task is presented to the agent and this relates to data coverage as well as
> b) characterizing when which method is useful
> will help bolster the work and should probably be included in a publication like this. As such, I will retain my score for now.

---

> ### Author Response · Authors · 2023-11-22
> **Comment to Reviewer 85iy after Final Revision**
>
> Thanks again for engaging candidly with our work. We have now uploaded our final revision which we believe addresses your two final queries.
>
> a) We’ve amended Section 2 again to explain exactly how tasks are inferred at test time and passed to the actor. We hope this is clear now. To answer your cheetah question directly, FB would expect to be provided a dataset of reward-labelled states at test time from the cheetah environment, where the rewards characterise the test task. Then, leveraging FB’s property that $z := \mathbb{E}_{(s, r) \sim \rho}[\mathcal{r} B(s)]$, we can infer the task using samples from the reward-labelled dataset. This task vector is then passed to the task-conditioned actor model.
>
> b) The most recent experiments carried out for 2wo9 provide better evidence of our claims in Section 5 that MC-FB performs worse than VC-FB because of poorer task coverage. (This was the testable hypothesis you referred to from Q5 in our original response to 2wo9). We lay out the new experiment and results at the start of Section 5.
>
> We hope you feel happy to update your score as a consequence. Thanks again for your feedback!

---

### Official Review · Reviewer_2wo9 · 2023-11-09

**Soundness:** 3 good
**Presentation:** 3 good
**Contribution:** 2 fair
**Rating:** 6
**Confidence:** 4

**Summary:**

The paper concerns itself with the OOD issue in zero-shot offline RL. It primarily makes CQL-like modifications to the objective for learning Forwad-Backward representations. From this arise two variants, the value-conservative variant which samples task representations uniformly and the measure conservative variant which uses the backward representation $B(s^+)$. Thorough experiments have been done on the ExORL benchmark where the method is compared to single-task RL (TD3, CQL) and non-conservative FB. It performs favorably in comparison to the baselines, sometimes even outperforming the single-task algorithms.

**Strengths:**

Addressing the issue of OOD actions arising  through maximization in offline RL is important, also in the zero-shot or multi-task case such as the case for FB.

Good experimental evaluation has been conducted.

I mostly didn't have trouble reading the paper, I think it's well-written.

The method is simple (application of CQL to FB) and therefore easy to understand.

**Weaknesses:**

Some parts of the theory are not clear, or maybe there were typos in the equations. The most essential thing for me is to clarify the treatment of the task representation vectors in  the VC and MC case and how do they connect to the original FB definition (see questions).

Why does VC outperform MC sometimes has been clarified to a certain extent in the discussion, however it is only intuition and no evidence for the claims have been given.

**Questions:**

Would be useful if you would properly define the abbreviations for VC-FB and MC-FB in the text before using them.

In equation 5  the $z$  variable is not an argument to the Q function, as it should be?

Broken english in line before equation 6. What is the connection between (5) and (6)? In (6) the forward representaiton here is a function of the same $z$, i.e. $\langle F(.., z), z \rangle$, however this is not how the Q value is defined in (5). Later you sample z's independently from $s^+$, hence the backward representation is completely independent of the task $z$ in the MC-FB. How is this valid? My questions here are:

* In which cases does the $z$ correspond to the backward representation?
* Follow up to the previous question, why does you MC-FB version  treat the backward representation and $z$ independently, while the VC-FB version does not, shouldn't the z in the F(...) be essentially the output of $B(s^+)$ as per your description?

The argument for the introduction of the MC-FB is  that it might not make sense to sample task vectors uniformly, but to focus on the ones that we care about (via backward rep), yet in figure 4 the zero shot performance of the MC variant is lower than VC? Can you clarify this? Also in the paper. (I realize that you have this in the discussion, however I think that it should be commented on earlier).

in 4.3, what does "stochastically dominate" mean?

Can you explain the failure cases of MC-FB in comparison to VC in figure 5?  For the Walker2D environment, the MC variant completely fails for the RND dataset - I suspect that the reason is low task coverage? Also, in some cases  there is a big gap between CQL and your method (random jaco).


If you address these concerns I will raise the score appropriately (the most important concern is the one about the dot-product).

---

> ### Author Response · Authors · 2023-11-14
> **Author Response to Reviewer 2wo9**
>
> We thank the reviewer for engaging with our manuscript and providing useful feedback. We summarise your comments/questions and respond to them below.
>
> **Q1: Would be useful if you would properly define the abbreviations for VC-FB and MC-FB in the text before using them.**
>
> Agreed. See Revision 4.1.
>
> **Q2: In equation 5 the $z$ variable is not an argument to the Q function, as it should be?**
>
> It is implicit via the superscript $\pi_z$, we took this notation directly from Touati et al (2022) but agree it could be clearer. We hope the rewrite of Section 2 proposed in Revision 2 will better clarify notation.
>
> **Q3: In which cases does the $z$ correspond to the backward representation?**
>
> As above, this question is valid, and a consequence of ambiguity in our introduction of the FB background theory. We hope to provide much better clarity via Revision 2.
>
> **Q4: Why don’t we use B(s_+) for z inside the forward model of MC-FB?**
>
> FB’s approximation of the successor measure **$M^{\pi_z}(s_0, a_0, s_+) \approx F(s_0, a_0, z)^\top B(s_+)$** is the cumulative time spent in future state $s_+$ starting in state $s_0$, taking action $a_0$ and attempting to solve task $z$. Our goal was to mitigate OOD successor measure overestimation *for all tasks* and so sample $z \sim \mathcal{Z}$ for input to the forward model. Our language justifying MC-FB was therefore imprecise, particularly the phrase: *“Instead, it may prove better to direct updates towards tasks we are likely to encounter*” (Section 3, paragraph 4). We agree with the reviewer that the best way to direct updates would be to use $z = B(s_+)$ as input to the forward model and, although not mentioned by the reviewer, as inputs to the Q functions for policy training (Equation 7). We believe it would no longer make sense to call such a variant _measure-conservative_, but rather _directed-value-conservative_, with all $z$s _directed_ by the backward embedding rather than sampled uniformly. It is straightforward to explore the usefulness of such a change, and we are running experiments on all datasets and domains to do so—see Revision 3. We will update the language used to justify MC-FB—see Revision 4.9.
>
> **Q5: (Our summary) Can you explain the poorer performance of MC-FB w.r.t VC-FB in aggregate, on walker RND, and on random jaco?**
>
> We suspect instances where MC-FB performance < VC-FB are explained by $z = B(s_+)$ providing worse coverage than $z \sim \mathcal{Z}$. This is a testable hypothesis that our evaluation of *directed* VC-FB (Revision 3) should help answer. If $z = B(s_+)$ does indeed provide poorer task coverage on some dataset/domain pairs, then _directed_ VC-FB should perform worse than MC-FB and VC-FB on those dataset/domain pairs.
>
> **Q6: In 4.3, what does "stochastically dominate" mean?**
>
> We take this term directly from Agarwal et. al (2021) where performance profiles are introduced. Formally, a random variable $M$ stochastically dominates $N$ if $P(M > x) \geq P(N > x) \; \forall x$. In practice the y-axis values of one algorithm’s performance profile are higher than another for all x-axis values. We will add a footnote on this, see Revision 4.2.

---

> > ### Author Response · Authors · 2023-11-20
> > **Comment to Reviewer 2wo9 after Revision One**
> >
> > We are commenting to update you on the progress of R3 (directed VC-FB). We are awaiting the completion of the training runs which are extensive. We thank you for your patience and will provide the results of the experiments as soon as they are ready. We hope that all of your other queries have been addressed in the updated manuscript or in the comment above. Please do let us know if you have any further questions!

---

> > > ### Comment · Reviewer_2wo9 · 2023-11-21
> > > **Read your response**
> > >
> > > Thank you for acknowledging my points, looking forward to see the results of the runs.

---

> > > > ### Author Response · Authors · 2023-11-22
> > > > **Comment to Reviewer 2wo9 after Final Revision**
> > > >
> > > > We have now added experimental evaluation of *directed* VC-FB to our final revision as per proposed revision R3. We report the new Table 2 here for convenience.
> > > >
> > > > | Dataset | Domain | Task | FB | DVC-FB | MC-FB | VC-FB |
> > > > | --- | --- | --- | --- | --- | --- | --- |
> > > > | All (100k) | all domains | all tasks | 99 | 108 | 136 | 148 |
> > > >
> > > > To recap: VC-FB regularises OOD actions on $F(s, a, z)^\top z$, with $s \sim \mathcal{D}$, and $z \sim \mathcal{Z}$; MC-FB regularises OOD actions on $F(s, a, z)^\top B(s_+)$, with $(s, s_+) \sim \mathcal{D}$ and $z \sim \mathcal{Z}$; and *directed*-VC-FB regularises OOD actions on $F(s, a, B(s_+))^\top B(s_+)$, with $(s, s_+) \sim \mathcal{D}$.
> > > >
> > > > As expected, if $|\mathcal{D}|$ is small, then deriving tasks from $B(s_+)$ with $s_+ \sim \mathcal{D}$ seems to provide poorer task coverage than $z \sim \mathcal{Z}$, and the extent to which $z$s are obtained in this way for each method correlates negatively with performance i.e. VC-FB > MC-FB > DVC-FB. We refer the reviewer to Appendix B.1.6 for implementation details, Appendix C for the full results and section 5 for further discussion. We hope this answers your original question, and sheds further light on discrepancies between VC-FB and MC-FB performance.
> > > >
> > > > Having responded to each of your concerns either experimentally or in writing, we hope you feel comfortable updating your score. Thanks very much again for your feedback.

---

### Author Response · Authors · 2023-11-14
**Global Response to All Reviewers (1/2)**

We thank all reviewers for engaging with our manuscript and providing valuable feedback. We use this global response for three purposes: 1) to summarise strengths highlighted by more than one reviewer, 2) to summarise weaknesses highlighted by more than one reviewer, and 3) to outline proposed revisions. We also respond to each reviewer individually.

### **Summary of Strengths**

- All reviewers agree that we provide a thorough and convincing experimental evaluation of our proposals;
- Reviewers 2wo9, 85iy, and xRo3 agree the problem setting is important, and that our proposed solution is well-justified and simple to implement;
- Reviewers xRo3, 85iy and 2wo9 agree our contributions are novel;
- Reviewers xRo3, 85iy and 2wo9 agree the paper is well-written and easy to follow.

### **Summary of Weaknesses**

*W1: Background (Section 2) Clarifications*

Reviewers 2wo9 and 85iy provided useful feedback on ambiguity in Section 2 which include, but are not limited to: task sampling distribution $\mathcal{Z}$, reordering of references, the relationship between task vectors and reward functions, and in what situations $z \sim \mathcal{Z}$ or $z = B(s)$. We plan to rewrite Section 2 to address each of these concerns--see Revision 2.

*W2: Performance on Random (100k) dataset(s)*

Reviewers xRo3 and 85iy questioned: 1) why our proposals perform relatively poorer on the low-quality 100k Random dataset than on the high-quality 100k RND dataset , and 2) how our proposals would perform on the full Random datasets. In response to 1) we hypothesised that the 100k Random dataset was fundamentally uninformative (penultimate sentence of Section 4.3, Q2) and therefore tricky to use for model/policy learning. The low absolute scores in Table 5 (Appendix C)  and in the visualisation on Maze in Appendix A.2 lend credence to this thesis. A response to question 2) may provide further evidence to this end. Should our proposals outperform the baselines on the full Random datasets then we are likely correct that the 100k variants are uninformative. We propose additional experiments to explore this, see revision Revision 1.

*W3: World Model Terminology*

Reviewers xRo3 and 5Tki questioned our use of *World Model* to describe (conservative) FB representations. The term has been used liberally in the literature to describe environment models used by agents to solve downstream tasks [1, 2, 3, 4, 5]. In our opinion, there is no agreement on what constitutes a *World Model* precisely, but we believe (conservative) FB representations (and the wider family of successor feature-based algorithms) capture the essence of Ha and Schmidhuber’s description as we explain in the opening paragraph of Section 1. Nonetheless, if the reviewer’s feel strongly that the term is being used inappropriately we would be happy to retitle the paper and drop the terminology. We note other authors utilising FB representations call them Behaviour Foundation Models (BFM) [6]. See Revision 5 for a proposed retitle.

---

> ### Author Response · Authors · 2023-11-14
> **Global Response to All Reviewers (2/2)**
>
> ### **Proposed Revisions**
>
> R1: Experimental evaluation of all FB methods on full Random and DIAYN datasets.
>
> R2: Rewriting of FB background theory in Section 2.
>
> R3: Experimental evaluation of a _directed_ variant of VC-FB with each $z \sim \mathcal{Z}$  replaced with $z = B(s_+)$. Experiments will be performed on all 100k datasets and domains.
>
> R4: Writing
>
> - R4.1 Introduce VC-FB and MC-FB abbreviations in the introduction.
> - R4.2 Footnote on stochastic dominance.
> - R4.3 Reference to Appendix B.1.2 on $\mathcal{Z}$ in the first sentence of the final paragraph of Section 2.
> - R4.4 Detail on how results are aggregated, that the score on a task is the cumulative episodic reward, and comparative comments on absolute scores Section 4.3. Removal of $x$ from Figure 4.
> - R4.5 Reviewer 85iy’s minor textual clarity suggestions.
> - R4.6 More related work on conservatism in offline RL.
> - R4.7 Clearer reference to Touati et al (2022) as origin of notation and derivation.
> - R4.8 Text speculating on when the zero-shot RL methods outperform single-task counterparts.
> - R4.9 Updated language justifying MC-FB in Section 3, Paragraph 4.
> - R4.10 Chagne Touati et al (2022) (preprint) citation to [7] (2023 ICLR submission).
>
> R5: Retitle work _Conservative Behaviour Foundation Models_, and drop _World Model_ terminology in text.
>
> R6: New appendix that derives losses for training *Conservative Successor Features* as our methods are fully compatible with non-FB zero-shot RL methods.
>
> We will provide an updated manuscript with these changes ASAP.
>
> ### **References**
>
> [1] LeCun, Yann. "A path towards autonomous machine intelligence version 0.9. 2, 2022-06-27." *Open Review* 62 (2022).
>
> [2] Chen, Chang, et al. "Transdreamer: Reinforcement learning with transformer world models." *arXiv preprint arXiv:2202.09481* (2022).
>
> [3] Xu, Yingchen, et al. "Learning General World Models in a Handful of Reward-Free Deployments." *Advances in Neural Information Processing Systems* 35 (2022): 26820-26838.
>
> [4] Hao, Shibo, et al. "Reasoning with language model is planning with world model." *arXiv preprint arXiv:2305.14992* (2023).
>
> [5] Kipf, Thomas, Elise Van der Pol, and Max Welling. "Contrastive learning of structured world models." *arXiv preprint arXiv:1911.12247* (2019).
>
> [6] https://openreview.net/forum?id=qnWtw3l0jb
>
> [7] Ahmed Touati, Jeremy Rapin, and Yann Ollivier. Does zero-shot reinforcement learning exist? In The Eleventh International Conference on Learning Representations, 2023

---

### Author Response · Authors · 2023-11-20
**Revision One**

In accordance with our list of planned edits, we have uploaded version 1 of our revised manuscript. We have made all planned revisions with the exception of R3 which will follow in the next two days. In addition to the planned revisions, we also provide an additional zero-shot RL baseline (successor features (SF) with laplacian eigenfunction features) in response to requests by Reviewer xRo3 and 5Tki. The baseline is the state-of-the-art SF-based zero-shot RL method as described in Touati et. al (2023). As planned, the title has been changed, but to a different title than first proposed in Global Response 2 as per discussions with reviewer 85iy. All revisions are colored blue.

We shall follow up with each reviewer individually with further comments, and shall provide version 2 ASAP. Thank you for your ongoing feedback.

---

### Author Response · Authors · 2023-11-22
**Final Revision**

We are writing to inform all reviewers we have uploaded our final revision. We have accomplished all we set out to achieve in our _Global Response to All Reviewers (2/2)_, and added a further baseline: SF with laplacian eigenfunctions. As before all revisions are colored blue.

We believe that all pressing concerns raised by reviewers have either a) been resolved via comment, or b) been answered with new experiments or clarifying text in the updated manuscript. We kindly ask that the reviewers reengage with the manuscript and update their scores to reflect changes and discussion since original reviews. Finally, we thank you once again for helping to improve this work.

---

### Meta-Review · Area_Chair_YzH3 · 2023-12-09

**Metareview:**

This paper studies the problem of zero-shot reinforcement learning, where the goal is to learn a representation that can be used to solve tasks without access to the original environment. The proposed approach builds on forward-backward representations, which aim to estimate the time-discounted probability of staying in a region of the state space. This representation can be used to compute the reward for future tasks, thereby enabling zero-shot policy optimization.

The reviewers generally agreed that the problem being studied is an important one. There was some concern about the limited novelty of the approach. Conservative penalties for offline reinforcement learning have been extensively studied in recent years. While combining them with forward-backward representations is somewhat incremental, such an approach is still valuable if it can be convincingly demonstrated that they are highly effective. However, there were still some remaining concerns about the clarity of the paper, especially in terms of the experiments. One particular concern was the introduction of two proposed methods; to be convincing, the authors need to propose a mechanism to choose which method to use in a zero-shot manner.

**Justification For Why Not Higher Score:**

There were remaining concerns from several reviewers about the experimental evaluation as well as the limited novelty.

**Justification For Why Not Lower Score:**

N/A

---

### Decision · Program_Chairs · 2024-01-16

Reject